# BLTP3A is associated with membranes of the late endocytic pathway and is an effector of CASM

Michael G Hanna [ID][1,2,3,4,5✉], Hely O Rodriguez Cruz [ID][1,2,3,4,5], Kenshiro Fujise[1,2,3,4,5], Yumei Wu[1,2,3,4,5], C Shan Xu[6], Song Pang[7], Zhuonging Li[8], Mara Monetti [ID][8] & Pietro De Camilli [ID][1,2,3,4,5✉]

## Abstract

Recent studies have identified a family of rod-shaped proteins thought to mediate lipid transfer at intracellular membrane contacts by a bridge-like mechanism. We show that one such protein, bridge-like lipid transfer protein 3A (BLTP3A)/UHRF1BP1 binds VAMP7 vesicles via its C-terminal region, and anchors them to lysosomes via its chorein domain-containing N-terminal region binding to Rab7. Upon lysosome damage, BLTP3A-positive vesicles rapidly (within minutes) dissociate from lysosomes. Lysosome damage is known to activate the CASM (Conjugation of ATG8 to Single Membranes) pathway, leading to lipidation and lysosomal recruitment of mammalian ATG8 (mATG8) proteins. We find that this process drives the reassociation of BLTP3A with damaged lysosomes via an interaction of its LIR motif with mATG8 which coincides with a dissociation from the vesicles. Our findings reveal that BLTP3A is an effector of CASM, potentially as part of a mechanism to help repair or minimize lysosome damage.

**Keywords** BLTP3; Lysosome; LC3; Rab45; Urate Crystals
**Subject Categories** Membranes & Trafficking; Organelles

## Introduction

Eukaryotic cells are populated by anatomically discontinuous lipid-based intracellular membranes and thus require mechanisms to transfer lipids between them. This is achieved both by membrane traffic (Palade, 1975) and by lipid transfer proteins (LTPs) (DeGrella and Simoni, 1982; Wirtz, 1991; Reinisch et al, 2024) or protein complexes that have the property to extract lipid from membranes, shield them in hydrophobic cavities and insert them into acceptor membranes. Typically, LTPs function at sites where two membranes are closely apposed, i.e. where transfer may occur with greater speed and specificity (Saheki and Camilli, 2017; Cohen et al, 2018; Scorrano et al, 2019; Wong et al, 2019; Prinz et al, 2020;

Voeltz et al, 2024). Moreover, in most cases studied so far transfer occurs via a shuttle mechanism in which a lipid harboring module that contains one or few lipids, is connected via flexible linkers to protein domains that tether the two membranes together (Prinz et al, 2020; Reinisch et al, 2024). In recent years, however, the occurrence of an additional mode of lipid transfer, mediated by rod-like proteins that harbor a hydrophobic groove or tunnel along which lipids can slide and which directly bridge two membranes has been described (Kumar et al, 2018; Osawa et al, 2019; Maeda et al, 2019; Valverde et al, 2019; Wong et al, 2019; Li et al, 2020; Leonzino et al, 2021; Dziurdzik and Conibear, 2021; Cai et al, 2022; Hanna et al, 2022; Adlakha et al, 2022; Kang et al, 2025; Banerjee et al, 2025). These proteins, collectively referred to as bridge-like lipid transfer proteins (BLTPs), are evolutionarily related and have a similar basic molecular architecture (Leonzino et al, 2021; Neuman et al, 2022; Levine, 2022; Hanna et al, 2023). Their core is represented by concatemers of small beta-sheets with a taco-like fold, referred to as repeating beta-groove (RBG) modules (Levine, 2022; Hanna et al, 2023), which are lined by hydrophobic amino acids (a.a.) at their inner surface, thus generating a continuous hydrophobic surface (Kumar et al, 2018; Li et al, 2020; Levine, 2022; Neuman et al, 2022; Hanna et al, 2023; Kang et al, 2025; Wang et al, 2025). Moreover, they comprise motifs or domains that allow them to tether two different membranes and a variety of loops or outpocketings of variable length which may have regulatory or protein-protein interaction functions (Kumar et al, 2018; Dziurdzik and Conibear, 2021; Guillén-Samander et al, 2021; Neuman et al, 2022; Adlakha et al, 2022; Guillén-Samander et al, 2022; van Vliet et al, 2022; Hanna et al, 2023; Kang et al, 2025; Wang et al, 2024, 2025). They are thought to mediate bulk lipid (generally phospholipid) transfer between membranes. As the groove at its narrowest point can only accommodate one phospholipid, such transport is thought to be unidirectional (Li et al, 2020; Kang et al, 2025; Wang et al, 2024). BLTPs comprise VPS13, the founding member of the family, as well as the autophagy factor ATG2, and other proteins originally referred to by multiple different names in different organisms that are now renamed BLTP1, BLTP2 and BLTP3 (Neuman et al, 2022; Hanna et al, 2023). These proteins differ in the number of RBG modules and thus in length: 17 in BLTP1, the longer member

[1]Department of Neuroscience, Yale University School of Medicine, New Haven, CT, USA. [2]Department of Cell Biology, Yale University School of Medicine, New Haven, CT, USA. [3]Howard Hughes Medical Institute, Yale University School of Medicine, New Haven, CT, USA. [4]Program in Cellular Neuroscience, Neurodegeneration and Repair, Yale University School of Medicine, New Haven, CT, USA. [5]Aligning Science Across Parkinson's (ASAP) Collaborative Research Network, Chevy Chase, MD, USA. [6]Department of Cellular and Molecular Physiology, Yale University School of Medicine, New Haven, CT, USA. [7]Yale University School of Medicine, New Haven, CT, USA. [8]Proteomics Core Facility, Sloan Kettering Institute, Memorial Sloan Kettering Cancer Center, New York, NY, USA. ✉E-mail: michael.hanna@yale.edu; pietro.decamilli@yale.edu

of the family and six in BLTP3, the shorter family member (Levine, 2022).

The putative role of BLTPs in bulk lipid transfer is ideally suited for membrane bilayer expansion or repair via the delivery of newly synthesized lipids from the ER, a possibility strongly supported by the well-established roles of yeast VPS13 in the growth of the sporulation membrane (Park and Neiman, 2012) and of ATG2 in the expansion of the isolation membrane (Wang et al, 2001; Velikkakath et al, 2012; Gómez-Sánchez et al, 2018; Osawa et al, 2019; Valverde et al, 2019). Other processes in which BLTPs anchored to the ER have been implicated also involve membrane expansion, such as biogenesis of mitochondria and of peroxisomes (Park et al, 2016; John Peter et al, 2017; Anding et al, 2018; Baldwin et al, 2021; Guillén-Samander et al, 2021), organelles not connected to the ER by membrane traffic. In other cases, the relation of the putative bulk lipid transfer function of BLTPs to bilayer expansion is less clear, and BLTPs seem to be primarily important to control the composition of the receiving bilayer (Tokai et al, 2000; John Peter et al, 2022; Wang et al, 2022; Hanna et al, 2022; Banerjee et al, 2025). In this case, delivery of new lipids to a membrane may be removed by a compensatory mechanism, for example, by membrane traffic, thereby limiting expansion overall. As BLTPs have been identified only recently, much remains to be discovered about their function.

Two very similar BLTPs of which little is known are BLTP3A (also called UHRF1BP1) and BLTP3B (also called SHIP164 or UHRF1BP1L for UHRF1BP1-Like) (Hanna et al, 2022; Neuman et al, 2022). BLTP3A was originally identified as a Binding Protein (BP) of the epigenetic regulator UHRF1 (Unoki et al, 2004) while its paralogue BLTP3B was independently identified as an interactor of syntaxin 6 (Syntaxin 6 Habc-interacting protein of 164 kDa, hence its alias SHIP164) and found to localize on membranes of the endocytic pathway harboring this SNARE protein (Otto et al, 2010). BLTP3A and BLTP3B were also top hits in a screen for effectors of Rab7 and Rab5, respectively (Gillingham et al, 2019).

Following up on these earlier studies, we have recently reported a systematic characterization of the properties of BLTP3B (Hanna et al, 2022). We showed that endogenous BLTP3B is localized on clusters of endocytic vesicles that interact with components of the retrograde microtubule-based transport system [dynein light chain (DYNLL1/2)] (Carter et al, 2016) and Rab45 (RASEF) (Wang et al, 2019) and that loss of BLTP3B results in a perturbation of the retrograde traffic to the Golgi area of the cation independent mannose-6-phosphate receptor (MPR) (Lin et al, 2003). Exogenous expression of BLTP3B, leading to its overexpression, resulted in a striking accumulation of these vesicles that formed tightly packed clusters anchored to Rab5-positive early endosomes (Hanna et al, 2022). How the putative lipid transfer function of BLTP3B relates to this localization and phenotypes remains an open question.

The goal of the present study was to acquire new information about BLTP3A, a risk gene for lupus erythematosus (Gateva et al, 2009; Zhang et al, 2011; Wen et al, 2020). We report here that BLTP3A, like BLTP3B, is localized on clusters of small vesicles and that its overexpression induces a massive expansion of such clusters. However, while clusters of BLTP3B-positive vesicles are anchored to Rab5-positive organelles, clusters of BLTP3A-positive vesicles are anchored to LAMP1-positive organelles via Rab7. We further show that lysosome damage triggers the rapid loss of the Rab7-dependent association of BLTP3A with LAMP1-positive

organelles, followed by its CASM (Durgan and Florey, 2022)-dependent reassociation with them in an mATG8 and LIR motif-dependent way. These findings have implications for lysosome health, as perturbation of their membranes reveals a faster lysis in BLTP3A KO cells than in control cells. Collectively, our results point to a role of this protein at the interface between late endocytic traffic and lysosomes and also raise the possibility that BLTP3A may play a role in the response to lysosome damage.

# Results

## Close structural similarity, but different interactions, of BLTP3A relative to BLTP3B

BLTP3A is very similar to BLTP3B (41% identity and 58% positives in primary sequence). Moreover, fold-prediction algorithms (Jumper et al, 2021; Yang et al, 2020) show that BLTP3A shares all the key structural features of BLTP3B: a rod-like core composed by six RBG motifs (Levine, 2022), a hydrophobic groove that runs along its entire length, a large disordered outpocketing of the rod-like core (a.a. 885–1188) and a C-terminal helix (a.a. 1394–1440) (Fig. 1A). Both BLTP3A and BLTP3B are reported in Biogrid (https://thebiogrid.org) to be interactors of Rab45, an adaptor for dynein and retrograde microtubule traffic (Wang et al, 2019) and, accordingly, over-expression of GFP-Rab45 in RPE-1 cells concentrated BLTP3A-mRFP to perinuclear spots (the centrosomal area) similar to our previous findings of BLTP3B (Fig. EV1A) (Hanna et al, 2022). Despite these similarities, BLTP3A lacks the motifs responsible for binding to syntaxin-6 (Stx6) and the dynein light chain (DYNLL1/2) (Fig. 1B), which we had identified in BLTP3B (Hanna et al, 2022). Moreover, BLTP3A was shown to be an effector of Rab7, instead of Rab5 (Gillingham et al, 2019). A per-residue evolutionary conservation analysis of BLTP3A carried out using the ConSurf server (Armon et al, 2001; Yariv et al, 2023) revealed that most conserved residues belong to the RBG motif core, particularly its N-terminal portion (Fig. 1A). Short stretches of conserved residues, however, are also present in predicted unfolded loops emerging from this core, including the large outpocketing. As revealed by western blotting, BLTP3A has broad expression in different mouse tissues (Fig. 1C), with higher levels occurring in brain and lung.

## BLTP3A localizes to foci concentrated near lysosomes

In a previous preliminary analysis of BLTP3A localization (Hanna et al, 2022), we had shown that exogenously expressed fluorescently tagged BLTP3A appears as fluorescent foci localized in proximity of organelles positive for the lysosome marker LAMP1 and of Rab7, a Rab associated with late endosomes and lysosomes (these organelles will be referred to henceforth collectively as "lysosomes"). As our study of BLTP3B had shown that its exogenous expression resulted in an enlargement of the BLTP3B-positive compartment (vesicle clusters), we wished to confirm that even endogenous BLTP3A was localized in proximity of lysosomes. Available antibodies directed against BLTP3A did not detect a clear specific signal by immunocytochemistry. Thus, we turned to a knock-in strategy to epitope tag the endogenous protein. Since preliminary experiments revealed heterogeneity in the expression

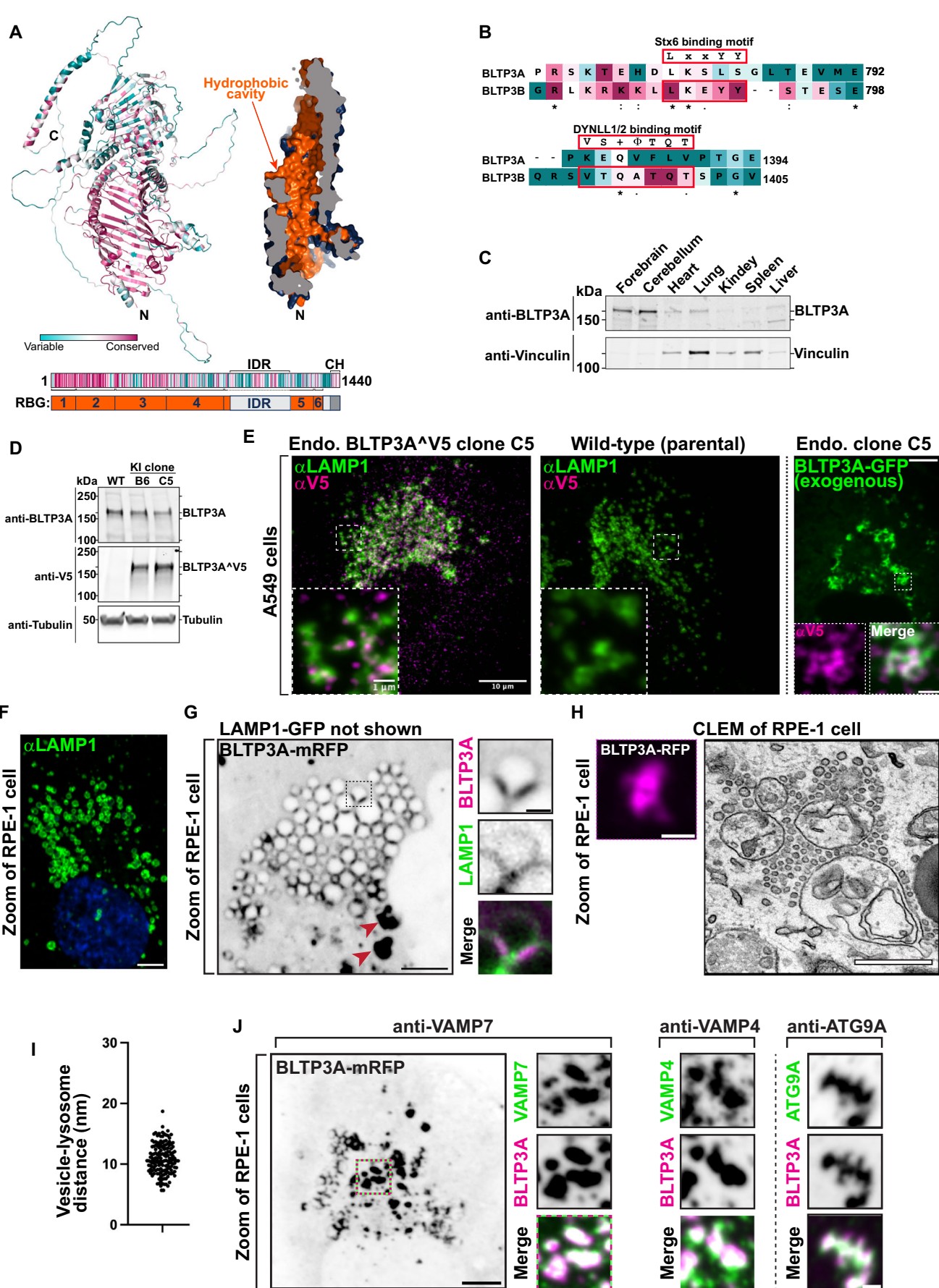

**Figure 1. Endogenous and exogenous BLTP3A form accumulations that associate with lysosomes.**

(A) AlphaFold prediction of full-length BLTP3A with ConSurf conservation scores (top left) for each a.a. or cross-section of surface rendering of BLTP3A channel (top right) highlighting hydrophobic residues (orange). Linear representation of BLTP3A: per residue ConSurf scores (top) and RBG organization (bottom). IDR predicted intrinsically disordered region (light gray), CH C-terminal helix (dark gray). (B) Alignment of the a.a. of motifs important for the indicated protein interactions of BLTP3B with corresponding sequences of BLTP3A. ConSurf conservation scores for each a.a. is indicated by color (same color scheme as in (A)). (C) Western blot of lysates of wild-type mouse tissues for BLTP3A and vinculin as a loading control. (D) Western blot of control and edited (BLTP3A^V5) cell clones for BLTP3A, V5, and alpha-tubulin as a loading control. (E) Fluorescence images of endogenously edited (left) or parental control (middle) A549 cells with antibodies against LAMP1 (green) and V5 (magenta). Scale bar, 10 μm. The insets are a zoom of a small region of the cell. Scale bar, 1 μm. Right: Fluorescence of exogenous BLTP3A-GFP (green) in endogenously edited A549 cell. Scale bar, 5 μm. Insets: zoom of square region of cell showing co-localization of endogenous BLTP3A^V5 signal from immunolabeling with antibodies against V5 (magenta) and BLTP3A-GFP fluorescence (green). Scale bar, 1 μm. (F) Fluorescence image of wild-type RPE-1 cell immunostained with antibodies against LAMP1 (green) and DAPI (blue). Scale bar, 5 μm. (G) Fluorescence image of an RPE-1 cell expressing exogenous BLTP3A-mRFP (large field, inverted grays) and LAMP1-GFP (not shown). Scale bar, 5 μm. The area enclosed by a dotted rectangle is shown at right at high magnification with BLTP3A-mRFP in magenta and LAMP1-GFP in green (individual channels are shown as inverted grays). Red arrows indicate large BLTP3A accumulations not associated with lysosomes. Scale bar, 1 μm. (H) CLEM of a BLTP3A-mRFP positive cluster in an RPE-1 cell. Left: fluorescence image of BLTP3A-mRFP (magenta). Scale bar, 1 μm. Right: EM micrograph of the field shown at left revealing that the BLTP3A-mRFP fluorescence reflects clusters of small vesicles, many of them tethered to the surface of lysosomes. Scale bar, 500 nm. (I) Distance between the membranes of lysosomes and tethered vesicles from EM micrographs. Mean = 10.8 nm; standard error of the mean = ±0.20 nm. (J) Left: Fluorescence image of an RPE-1 cell expressing exogenous BLTP3A-mRFP (inverted grays) and immunolabeled with antibodies against VAMP7 (shown in the high mag fields at right). Scale bar, 5 μm. Right: zooms of different RPE-1 cells expressing exogenous BLTP3A-mRFP (magenta) and immunolabeled (green) with antibodies against the following endogenous proteins, VAMP4 and ATG9A. Individual channels are shown as inverted grays. Merge of channels on bottom. Scale bar, 1 μm.

levels of BLTP3A in frequently used cell lines, we chose A549 cells (lung adenocarcinoma epithelial cells) for these experiments where we detected robust expression of BLTP3A, consistent with high expression of BLTP3A in the lung (Fig. 1C).

The BLTP3A locus in A549 cells was edited by inserting after residue V904 a nucleotide sequence encoding a single V5 epitope flanked on either side with short GSGSG linkers (Fig. EV1B). This site is within a predicted disordered region and is not expected to change the lipid channel core of BLTP3A (Fig. EV1C). Edited BLTP3A was validated by Western blotting of homogenates of edited cells revealing a V5-positive band (endogenous BLTP3A^V5) with the same motility as BLTP3A (Fig. 1D).

Anti-V5 immunofluorescence of edited cells showed small weakly fluorescence puncta throughout the cytoplasm, which were enriched in central regions of cells (Fig. 1E) and were not observed in WT cells immunostained under the same conditions (Fig. 1E). Importantly, many V5-positive puncta were adjacent to, or partially overlapping with, the fluorescence produced by antibodies against LAMP1 (Fig. 1E), consistent with BLTP3A being a Rab7 effector. The fluorescence of exogenously expressed tagged BLTP3A-GFP overlapped with anti-V5 immunofluorescence of edited cells, although accumulations of BLTP3A-GFP fluorescence were much larger (Fig. 1E).

## Large BLTP3A foci produced by BLTP3A overexpression represent clusters of vesicles anchored to lysosomes

Next we performed Correlative Light-Electron Microscopy (CLEM) to reveal the features of BLTP3A-positve structures. Given the similar localization of endogenous and exogenous BLTP3A next to lysosomes, we capitalized on exogenous BLTP3A tagged with various fluorescent proteins, i.e. constructs which could be analyzed by this technique (immunofluorescence of the V5 epitope requires fixation and permeabilization, i.e. a treatment that perturbs cell ultrastructure). We primarily used RPE-1 cells for these studies, as LAMP1-positive organelles in these cells are very large, abundant, clustered around the Golgi complex and nucleus (Fig. 1F), and thus easy to identify by microscopy even without specific markers. Upon co-expression of LAMP1-GFP and BLTP3A-mRFP (Fig. 1G), two

sets of BLTP3A-mRFP-positive structures were observed by fluorescence microscopy: (i) BLTP3A-mRFP accumulations directly adjacent to lysosomes, often bridging two closely apposed LAMP1-GFP-vacuoles and (ii) larger (often very large) BLTP3A-mRFP accumulations not obviously connected to the large LAMP1-GFP-vacuoles. Analysis of these structures by CLEM in RPE-1 cells also expressing mito-BFP (to help fluorescence - EM alignment) showed that BLTP3A-mRFP foci represented tightly packed clusters of ~50–70 nm vesicles (Fig. 1H). Importantly, the vesicles of such clusters directly adjacent to LAMP1-GFP vacuoles appeared to be tethered to lysosomal membranes, with an average distance of ~10–11 nm (Fig. 1I).

This exaggerated accumulation of large clusters of small vesicles mirrors what we had observed upon overexpression of BLTP3B, indicating that a shared property of BLTP3A and BLTP3B is to bind small vesicles, induce their accumulation, cluster them, and anchor such clusters to other organelles—although clusters of BLTP3B vesicles are anchored to early endosomes (consistent with BLTP3B being an effector of Rab5), while clusters of BLTP3A are anchored to late endosomes/lysosomes (consistent with BLTP3A being an effector of Rab7). As our study of BLTP3B had shown that even BLTP3B expressed at an endogenous level is localized to vesicle clusters which are much smaller than clusters observed upon BLTP3B overexpression (Hanna et al, 2022), we hypothesized that the massive accumulation of vesicles observed upon overexpression of BLTP3 isoforms reflect a property of these proteins to nucleate biomolecular condensates. Accordingly, live imaging revealed that both clusters of BLTP3A and of BLTP3B (see Video 1 and Fig. 8E from Hanna et al, 2022) are highly dynamic. For example, they can undergo fission into smaller clusters, as expected for a compartment with liquid-like properties.

## BLTP3A-positive vesicles contain VAMP7, a SNARE implicated in traffic to lysosomes

The clustering of BLTP3A-positive vesicles next to lysosomes suggested that they may represent organelles destined to fuse with them, perhaps arrested at a docking stage due to a dominant negative effect of BLTP3A overexpression. As at least some of the

vesicles that fuse with late endosomes and lysosomes harbor VAMP7 in their membrane (Advani et al, 1999; Pols et al, 2013), we explored the potential presence of this SNARE in BLTP3A-positive vesicles (Fig. 1J). Supporting this hypothesis, anti-VAMP7 immunofluorescence revealed a striking overlap with the fluorescence of BLTP3A-mRFP both on the isolated BLTP3A clusters and on those anchored to lysosomes (Fig. 1J). A similar overlap was observed between the BLTP3A signal and immunofluorescence for VAMP4 (Fig. 1J), another SNARE protein implicated in endosomal traffic (Martinez-Arca et al, 2001; Mallard et al, 2002; Tran et al, 2007). In spite of the many similarities between BLTP3A and BLTP3B, only minimal overlap was observed between BLTP3B-mRFP fluorescence and endogenous VAMP7 or VAMP4 immunoreactivity (Fig. EV1D), revealing differences in the cargo of BLTP3A and BLTP3B vesicles. However, BLTP3A-mRFP accumulations also overlapped with the immunofluorescence of endogenous ATG9A, a component of autophagosome precursor vesicles (Fig. 1J), as previously observed for BLTP3B-mRFP (Hanna et al, 2022) (Fig. EV1D).

## The association of BLTP3A with lysosomes is mediated by its N-terminal region where the Rab7 binding site is located

To confirm that Rab7 is responsible for the association of BLTP3A-positive vesicles with lysosomes (Gillingham et al, 2019), BLTP3A-mRFP was co-expressed in RPE-1 cells with either WT RAB7 (GFP-Rab7$_{WT}$) or dominant negative (DN) Rab7 (GFP-Rab7$_{T22N}$), i.e. a mutant Rab7 that sequesters its guanylnucleotide exchange factor (GEF) to prevent formation of GTP-loaded Rab7 (Fig. 2A) (Stenmark and Olkkonen, 2001). The co-expression of WT Rab7, which localized along the entire surface of endolysosomes, but not within the vesicle clusters, did not alter the localization of BLTP3A-mRFP (Fig. 2A). In contrast, co-expression of GFP-Rab7$_{T22N}$ abolished the association of BLTP3A-mRFP foci with lysosomes and induced the expansion of the BLTP3A foci free in the cytoplasm (Fig. 2A), which CLEM confirmed to represent large accumulations of vesicles no longer associated with lysosomes (Fig. 2B).

In order to determine the region of BLTP3A responsible for the association with Rab7, which could provide insight into the orientation of BLTP3A at the lysosome-vesicle interface, we generated chimeras of BLTP3A and BLTP3B using BLTP3B tagged with mRFP at its C terminus as a backbone (Fig. 2C). As BLTP3B does not associate with Rab7 in spite of its close similarity to BLTP3A, we searched for a.a. sequences of BLTP3A which would confer Rab7 binding and lysosome localization to BLTP3B. A chimera (BLTP3$_{chimera-1}$) in which its first RBG module (a.a. 1–125, which includes the so-called chorein domain) was replaced by the first RBG module of BLTP3A (a.a 1–125) formed large clusters but such clusters were not associated with lysosomes (Fig. 2D), as expected for BLTP3B. In contrast, a chimera (BLTP3$_{chimera-2}$) in which its second RBG module (a.a. 126–319) was replaced by the equivalent module of BLTP3A (a.a. 126–322) localized to LAMP1-GFP compartments similar to WT BLTP3A (Fig. 2D), suggesting that the second RBG module of BLTP3A is sufficient for the Rab7-dependent lysosomal localization.

Moreover, a truncated construct comprising the first 2 RBG motifs of BLTP3A plus the first β-strand of the third RBG module of the same protein resulted in a fusion protein (BLTP3A$_{1-336}$-mRFP) that localized at lysosomes (Fig. 2D,E), confirming the presence of the Rab7 binding site in this BLTP3A fragment. Notably, this fragment decorated homogenously the entire lysosomal surface, without forming the foci on their surface that reflect vesicle accumulations. These findings were further supported by the exogenous expression of Rab7 constructs. Expression of WT Rab7 greatly enhanced the localization of BLTP3A$_{1-336}$-mRFP around the entire lysosomal surface (Fig. 2F), while the expression of dominant negative Rab7 (GFP-Rab7$_{T22N}$) resulted in a diffuse localization of BLTP3A$_{1-336}$-mRFP throughout the cytosol (Fig. 2F). We conclude that a portion of BLTP3A near its N terminus is necessary and sufficient for the localization of BLTP3A at lysosomes.

## The C-terminal region of BLTP3A mediates its interaction with vesicles

Rab7-dependent binding of BLTP3A to lysosomes via its N-terminal region implies that its C-terminal portion is likely responsible for vesicle binding and clustering (Fig. 2G). To test this hypothesis, we generated BLTP3A constructs with C-terminal deletions and expressed them in cells also expressing dominant negative Rab7 (GFP-Rab7$_{T22N}$) to determine whether they could still cluster vesicles (Fig. 3A). Deletion of the C-terminal helix of BLTP3A and of the linker that connects this helix to the last RBG motif (BLTP3A$_{1-1364}$-mRFP) did not affect the property of BLTP3A to cluster VAMP7-vesicles (detected by immunofluorescence using antibodies against VAMP7) similar to wild-type BLTP3A. However, a further truncation (construct BLTP3A$_{1-1327}$-mRFP), including the last two beta-strands of the sixth and final RBG motif of the channel, resulted in a protein that was diffusely cytosolic and did not cluster vesicles. The only accumulation of BLTP3A observed in these cells was a single cluster close to the nucleus (Fig. 3A), most likely reflecting its interaction with Rab45 at the centrosomal area (Fig. EV1A), but this cluster was VAMP7 negative, in agreement with the loss of vesicle binding (Fig. 3A) Neither C-terminal truncation abolished localization to lysosomes (Fig. EV1E). We conclude that the C-terminal portion of BLTP3A is necessary to interface with small vesicles.

We also attempted to identify BLTP3A binding partners on the vesicles. Towards this aim, we carried out anti-V5 affinity purification from non-ionic detergent solubilized A549 cells where BLTP3A was tagged at the endogenous locus, i.e. experimental conditions optimally suited to reveal physiological binding partners (Fig. 3B). Affinity-purified proteins were then identified by mass spectrometry (Fig. 3C). Unedited A549 cells were used as controls. This search did not identify any obvious candidate binding protein. While two of the top specific hits, Rab27B and its effector melanophilin (MLPH) are membrane associated proteins of transport vesicles (Nagata et al, 1990; Ménasché et al, 2000; Hume et al, 2001; Bahadoran et al, 2001; Nagashima et al, 2002; Strom et al, 2002), we failed to obtain evidence for a concentration of these proteins on BLTP3A-positve vesicles. Thus these two proteins were not further studied in the context of the present study. Two other top hits, however, the two mATG8 family members GABARAP and MAP1LC3B, provided insight into properties of BLTP3A which will be discussed below.

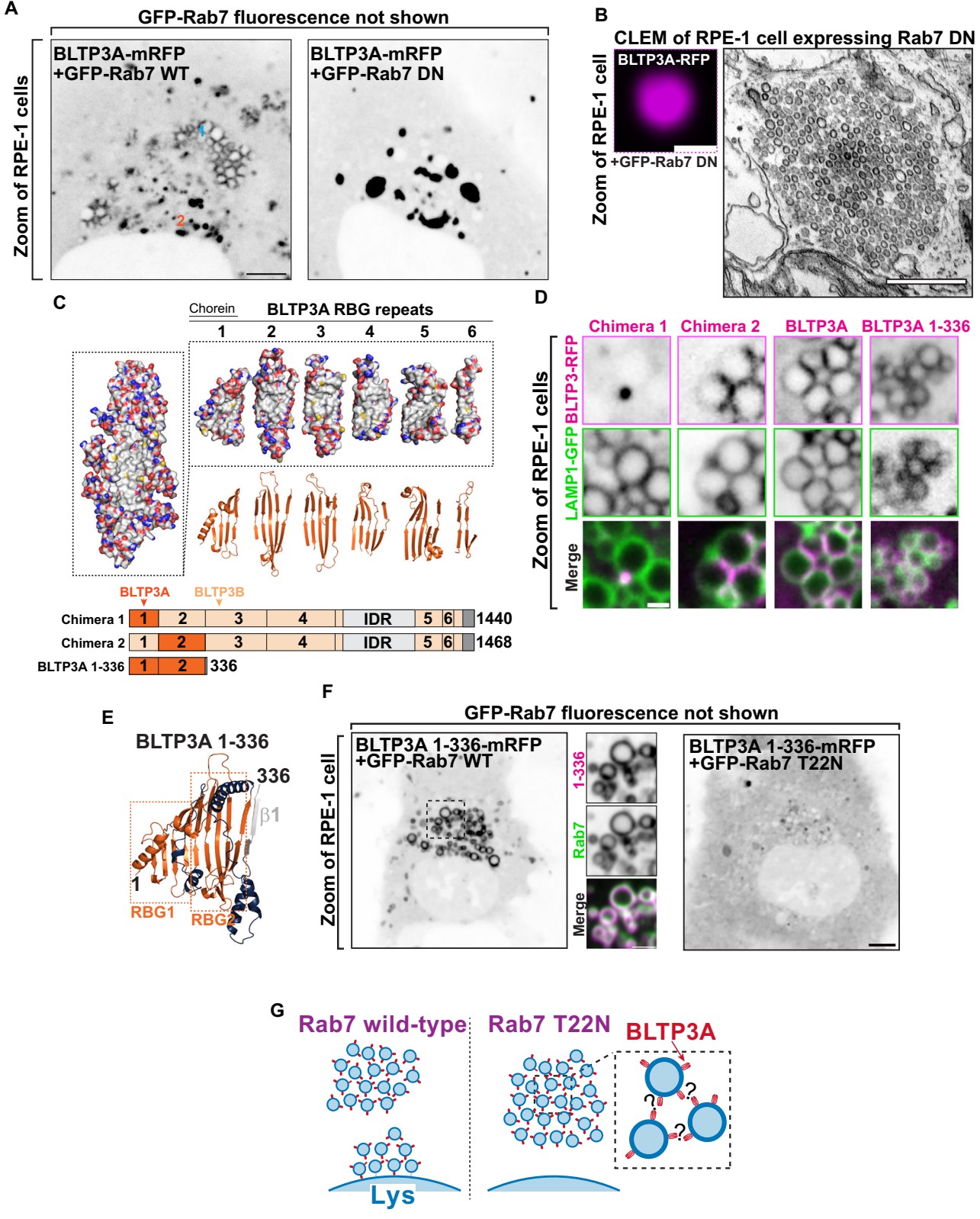

**Figure 2. The N terminus of BLTP3A associates with Rab7 on the surface of lysosomes.**

(**A**) Fluorescence images of RPE-1 cells expressing exogenous BLTP3A-mRFP (shown in inverted grays) and (not shown) GFP-tagged wild-type Rab7 (left) or dominant negative (DN) Rab7 (T22N) (right). Scale bar, 5 μm. (**B**) CLEM of a BLTP3A-mRFP positive cluster in an RPE-1 cell expressing GFP-tagged dominant negative Rab7. Left: fluorescence image of BLTP3A-mRFP (magenta). Scale bar, 1 μm. Right: EM micrograph of the field shown at left revealing that the BLTP3A-mRFP fluorescence reflects clusters of small vesicles. Scale bar, 500 nm. (**C**) BLTP3 chimeras design. Left: Surface representation of the predicted RBG core of BLTP3A. Red and blue indicate positive and negative charges, respectively, and gray indicates hydrophobic surfaces. Right: Surface representation (top) and ribbon representation (bottom) of the "untwisted" protein showing individual RBG motifs. Bottom: Cartoon of chimeras consisting of BLTP3A (dark orange) and BLTP3B (light orange) RBG motifs. (**D**) High-magnification live fluorescence images of RPE-1 cells expressing the indicated BLTP3-mRFP constructs (magenta) and LAMP1-GFP (green). Individual channels are shown as inverted grays. Scale bar, 1 μm. (**E**) Ribbon representation of the AlphaFold prediction of a.a. 1–336 of BLTP3A. Blue indicates loops connecting adjoining RBG motifs, and gray indicates the first beta-strand of the third RBG motif. (**F**) Fluorescence images (inverted grays) of RPE-1 cells expressing BLTP3A-1-336-mRFP and either (not shown) GFP-Rab7 (left), or GFP-Rab7 T22N (right). Scale bar, 5 μm. A zoom of an area of the cell at left (dotted square) expressing BLTP3A-1-336-mRFP (magenta) is also shown, along with the Rab7 fluorescence (green), demonstrating the localization of BLTP3A-1-336-mRFP around the entire profile of lysosomes. Individual channels are shown as inverted grays. Scale bar, 2 μm. (**G**) Cartoon depicting the proposed association of BLTP3A vesicle clusters with the surface of lysosomes and the dependence of this association on Rab7.

## Lysosomal damage disrupts the Rab7-dependent association of BLTP3A-positive vesicles with lysosomes

Perturbation of the membranes of lysosomes, for example by L-Leucyl-L-Leucine methyl ester (LLOMe), a dipeptide taken-up into lysosomes where it is metabolized into membranolytic peptides (Goldman and Kaplan, 1973; Thiele and Lipsky, 1990; Uchimoto et al, 1999), was reported to trigger the rapid recruitment of factors to their surface that may help prevent or repair damage (Skowyra et al, 2018; Radulovic et al, 2018; Shukla et al, 2022; Herbst et al, 2020; Radulovic et al, 2022; Tan and Finkel, 2022; Bentley-DeSousa and Ferguson, 2025; Wang et al, 2025). These include, besides ESCRT components, shuttle-like lipid transfer proteins such as ORP family members (Tan and Finkel, 2022; Radulovic et al, 2022) and bridge-like lipid transfer proteins structurally related to BLTP3A (Wong et al, 2019; Neuman et al, 2022) such as VPS13C (Wang et al, 2025) and ATG2 (Tan and Finkel, 2022; Cross et al, 2023). Both VPS13C and ATG2 are thought to mediate bulk phospholipid delivery to damaged lysosomes from the ER, which they bind via their N-terminal chorein domain (Kumar et al, 2018; Valverde et al, 2019; Osawa et al, 2019; Maeda et al, 2019; Wang et al, 2025).

Although a pool of BLTP3A is already present on lysosomes under control conditions (but in an opposite orientation compared to that of the related proteins ATG2 and VPS13C when bound to damaged lysosomes, i.e. with its N-terminal chorein domain facing the lysosomal membrane), we explored whether LLOMe dependent damage of lysosomes had an impact on BLTP3A localization in RPE-1 cells. Surprisingly, and in contrast to the damage-dependent recruitment of VPS13C (Wang et al, 2025) and ATG2 (Tan and Finkel, 2022) to lysosomes, the focal accumulations of BLTP3A on lysosomes, which reflect accumulations of BLTP3A-positive vesicles, dissociated within minutes from the lysosomal surface upon its damage (Fig. 4A; Movie EV1), whose occurrence was confirmed by the recruitment of cytosolic IST1 (mApple-IST1) (Fig. 4A; Movie EV1), an ESCRT-III subunit (Skowyra et al, 2018; Corkery et al, 2024). This LLOMe-dependent dissociation of BLTP3A-positive vesicle clusters from lysosomes, however, was not accompanied by a dispersion of the vesicle clusters themselves, indicating that the Rab7-dependent interaction of BLTP3A with lysosomes, not the interactions of BLTP3A which bind and clusters vesicles, was perturbed. This was confirmed by the finding that the Rab7 binding N-terminal fragment of BLTP3A (BLTP3A$_{1-336}$-mRFP), which does not bind vesicles, dissociated from lysosomes,

visualized in this experiment by the lysosomal protein NPC1 (NPC1-GFP) (Fig. 4B; Movie EV2). Mechanisms responsible for the dissociation of BLTP3A from lysosomes remain unknown. BLTP3A dissociation was not due to loss of Rab7 binding sites on lysosomes as the binding of VPS13C, which is recruited to lysosomes with a kinetic similar to that of BLTP3A dissociation, requires active Rab7 at these organelles (Wang et al, 2025). Dissociation was not due to competition by the newly recruited VPS13C because it also occurred in VPS13C KO cells (Fig. EV2A,B; Movies EV3 and 4). Likewise dissociation was not due to the rapid phosphorylation of Rab7 at serine 72 in response to LLOMe addition, a process primarily mediated by the kinase activity of LRRK1 (Wang et al, 2025) with an additional variable contribution of the kinase TBK1 (Nirujogi et al, 2021; Fujita et al, 2022; Heo et al, 2018; Talaia et al, 2024), as over-expression of a dominant-protein kinase active LRRK1 mutant (GFP-LRRK1$^{K746G}$) did not affect the localization of BLTP3A-mRFP foci next to lysosomes in RPE-1 cells (Fig. EV2C,D).

## CASM activation induces the reassociation of BLTP3A with lysosomal membranes

One event triggered by LLOMe-dependent damage of the lysosomal membrane is activation of CASM (Conjugation of Atg8 to Single Membranes) (Durgan and Florey, 2022; Boyle et al, 2023; Corkery et al, 2023; Kaur et al, 2023; Fischer et al, 2020). This is the process whereby lysosome perturbations that drive V-ATPase V0-V1 association in their membrane to enhance its proton pump activity also result in the recruitment of a subset of components of the classical autophagy pathway resulting in the lipidation and recruitment to the lysosomal membrane of mATG8 family proteins. These are small adaptors that are recruited to membranes in response to their triggered conjugation to PE or PS and bind proteins which contain the so-called LC3-interacting region (LIR) motif, typically found in disordered protein regions (Rogov et al, 2023). Atg8 family proteins, which comprise six members in mammals, are well established players in conventional autophagy (Melia et al, 2020; Nieto-Torres et al, 2021; Figueras-Novoa et al, 2024; Deretic et al, 2024): they interact with the isolation membrane via their lipid tail and recruit cargo targeted for autophagy via their LIR-motif-dependent interactions. However, the discovery of CASM has now revealed another important role of these proteins which is being intensely investigated (Durgan and Florey, 2022).

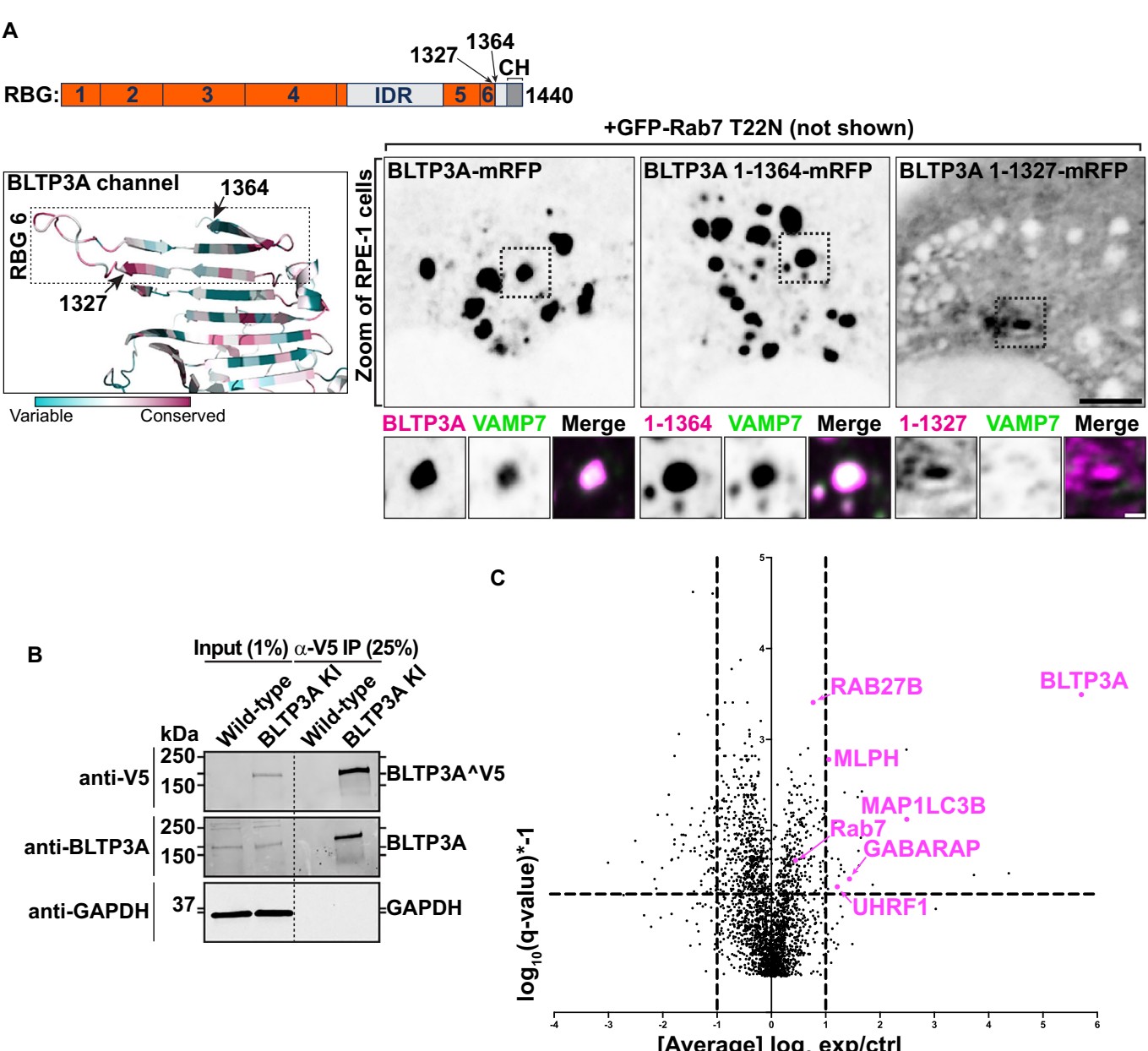

**Figure 3. The C terminus of BLTP3A associates with vesicles.**

(**A**) Top: Linear representation of BLTP3A RBG organization and C-terminal truncations indicated by arrows. Bottom left: AlphaFold-based structure of the C terminus of BLTP3A channel. Individual residues are colored by conservation scores as Fig. 1A. Truncations are indicated by arrows. Bottom right: Fluorescence images (inverted grays) of RPE-1 cells expressing the indicated BLTP3A-mRFP constructs along with dominant negative Rab7 (not shown) demonstrating that the property of BLTP3A to bind and cluster vesicles is dependent on its region comprised between a.a. 1327 and 1364. Note that the construct 1–1327, shows a focal accumulation next to the nucleus, which is VAMP7 negative, likely reflecting its pool bound to Rab45 (see Fig. EV1A). Scale bar, 5 μm. Zoomed images (dotted squares) are shown below the main field along with VAMP7 fluorescence. Scale bar, 1 μm. (**B**) Western blots of cell extracts (inputs) of control and edited A549 cells, and of material immunoisolated from these extracts by anti-V5 magnetic beads. Immunolabeling for BLTP3A, V5 (endogenously tagged BLTP3A), and for GAPDH as a loading control, are shown. (**C**) Scatter plot of mass spectrometry-identified proteins in immunoisolated material from either control or endogenously edited BLTP3A^V5 A549 cells using anti-V5 magnetic beads ($N = 3$, biological replicates). Proteins significantly enriched in material immunoisolated form edited cells compared to wild-type cells are plotted in the right-top quadrant. Proteins of note are labeled in magenta.

In view of the identification of two mATG8 proteins, MAP1LC3B and GABARAP, as interactors of BLTP3A (Fig. 3C), we further explored a potential role of CASM in BLTP3A dynamics. A search for LIR motifs in BLTP3A using the publicly available iLIR Autophagy Database (https://ilir.warwick.ac.uk/index.php)

predicts such a motif [Position Specific Scoring Matrix (PSSM) score: 16] within a disordered loop projecting out from the C-terminal region of BLTP3A, but not of BLTP3B, from several mammalian species, including humans (a.a. 1129–1134) (Fig. 5A) (see also (Tu and Brumell, 2020)). The high degree of conservation

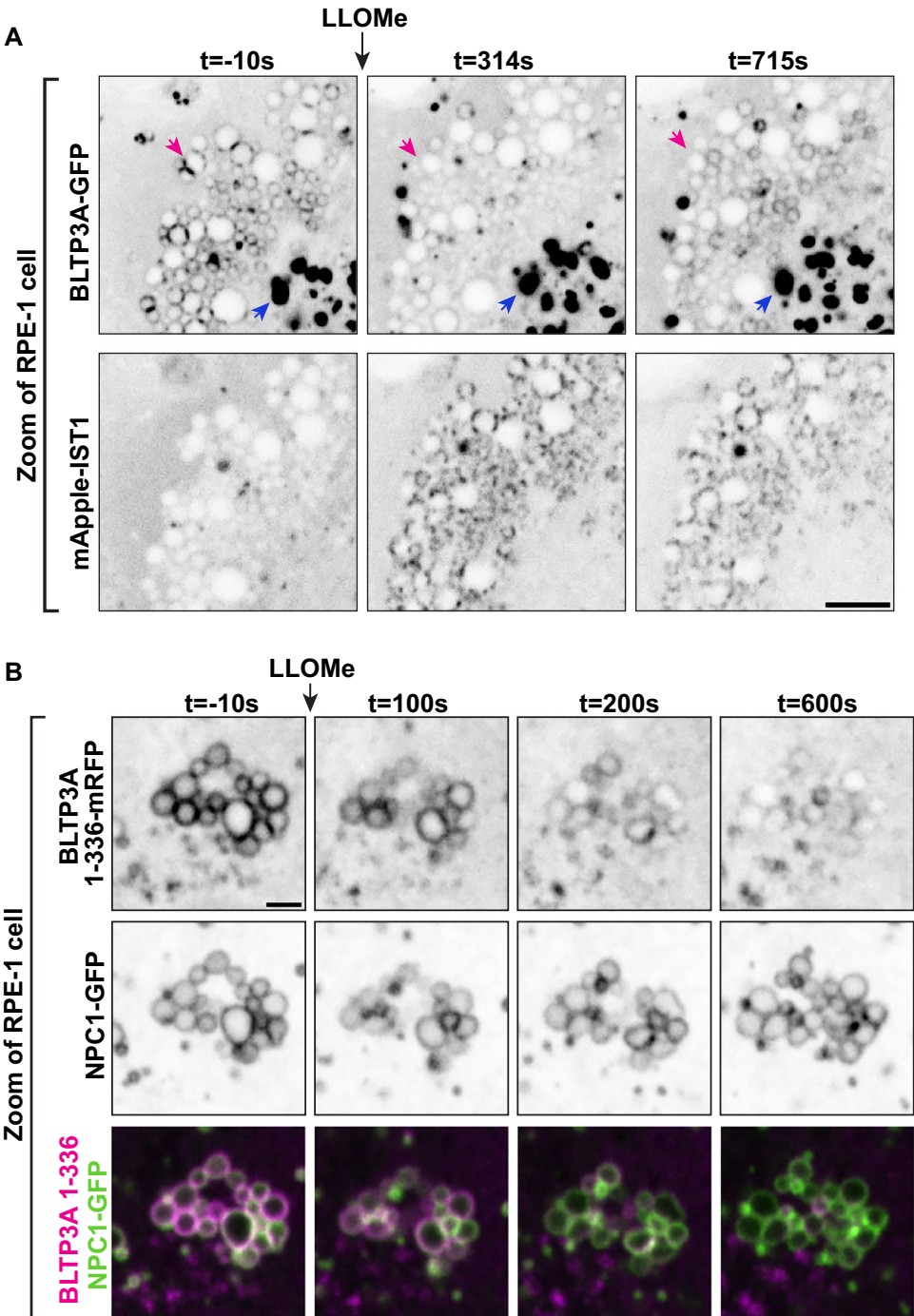

**Figure 4. Exogenous BLTP3A is shed from the surface of lysosomes upon damage of their membranes.**

(A) Time-series of live fluorescence images (inverted grays) of exogenous BLTP3A-GFP and mApple-IST1 before and after addition of LLOMe. Arrowheads (magenta) point to BLTP3A accumulations shed from lysosomes upon addition of LLOMe. Scale bar, 5 μm. (B) Time-series of live fluorescence images of BLTP3A-1-336-mRFP (magenta) and the lysosomal marker NPC1-GFP (green). Fluorescence of individual channels is shown in inverted grays. Scale bar, 2 μm.

of this motif relative to its surrounding a.a. sequences suggests its physiological importance, consistent with our co-affinity purification results. Moreover, structure-prediction algorithms (Abramson et al, 2024) predict with high confidence an interaction between the LIR motif of BLTP3A and the majority of the six known mATG8 proteins (Fig. EV3A,B).

To determine a potential physiological role of an mATG8-BLTP3A interaction, we co-expressed BLTP3A-mRFP and GFP-LC3B in RPE-1 cells. The localization of BLTP3A-mRFP on lysosomes and to large accumulations was not changed by the over-expression of GFP-LC3B, which was mostly cytosolic (Fig. 5B). Starvation of these cells resulted in the formation of GFP-LC3B-positive foci, as expected, but did not

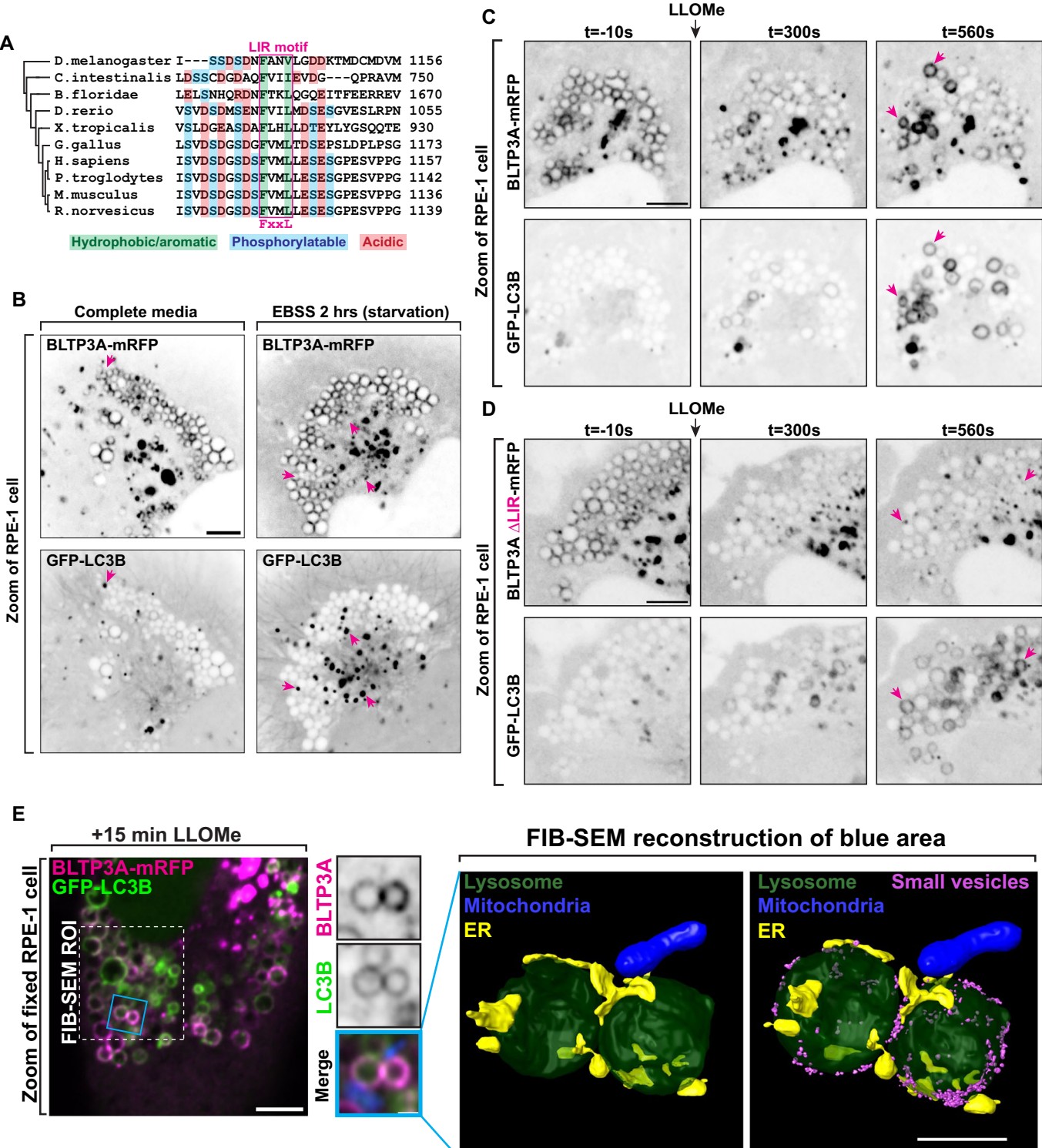

**A**

LIR motif

| | |
|---|---|
| D.melanogaster | I---SSDSDNFANVLGDDKTMDCMDVM 1156 |
| C.intestinalis | LDSSCDGDAQFVIIEVDG---QPRAVM 750 |
| B.floridae | LELSNHQRDNFTKLQGQEITFEERREV 1670 |
| D.rerio | VSVDSDMSENFVILMDSESGVESLRPN 1055 |
| X.tropicalis | VSLDGEASDAFLHLLDTEYLYGSQQTE 930 |
| G.gallus | ISVDSDGSDSGFVMLIDSEPSLDPLPSG 1173 |
| H.sapiens | ISVDSDGSDSFVMLLESESGPESVPPG 1157 |
| P.troglodytes | ISVDSDGSDSFVMLLESESGPESVPPG 1142 |
| M.musculus | ISVDSDGSDSFVMLLESESGPESVPPG 1136 |
| R.norvesicus | ISVDSDGSDSFVMLLESESGPESVPPG 1139 |

FxxL

Hydrophobic/aromatic    Phosphorylatable    Acidic

**B** — Complete media / EBSS 2 hrs (starvation); Zoom of RPE-1 cell; BLTP3A-mRFP, GFP-LC3B

**C** — LLOMe; Zoom of RPE-1 cell; BLTP3A-mRFP, GFP-LC3B; t=−10s, t=300s, t=560s

**D** — LLOMe; Zoom of RPE-1 cell; BLTP3A ΔLIR-mRFP, GFP-LC3B; t=−10s, t=300s, t=560s

**E** — +15 min LLOMe; Zoom of fixed RPE-1 cell; BLTP3A-mRFP, GFP-LC3B; FIB-SEM ROI; BLTP3A, LC3B, Merge

**FIB-SEM reconstruction of blue area**

Lysosome, Mitochondria, ER — Lysosome, Mitochondria, ER, Small vesicles

alter the localization of BLTP3A-mRFP, indicating that BLTP3A does not play a role in conventional autophagy (Fig. 5B).

We next monitored the response of LC3B relative to BLTP3A after addition of LLOMe. In the first few minutes after LLOMe addition, when BLTP3A-mRFP clusters as described above dissociated from lysosomes, GFP-LC3B fluorescence remained cytosolic with no overlap with the BLTP3A fluorescence (Fig. 5C; Movie EV5). After ~5–10 min of LLOMe treatment, however, GFP-LC3B began to accumulate on the surface of some lysosomes (Fig. 5C; Movie EV5), as previously reported (Cross et al, 2023), and this association correlated with the reassociation of a pool of BLTP3A-mRFP to such lysosomes, consistent with BLTP3A being

an mATG8 effector. This BLTP3A pool colocalized with LC3 along the entire surface of lysosomes and did not occur in focal accumulations indicating that it did not reflect presence of large vesicle clusters, although large BLTP3A-positive clusters persisted in the surrounding cytoplasm (Figs. 4A and 5C). Similar results were observed upon addition of the lysosome stressor glycyl-L-phenylalanine 2-naphthylamide (GPN) (Fig. EV3C; Movie EV6) (Chen et al, 2024; Durgan and Florey, 2022), another CASM activator. Deletion of the LIR motif of BLTP3A (BLTP3AΔLIR-mRFP) did not affect the localization of BLTP3A in the absence of LLOMe treatment and did not abolish the shedding of BLTP3A clusters upon LLOMe treatment but abolished its recruitment to GFP-LC3B positive lysosomes after LLOMe (Fig. 5D; Movie EV7), demonstrating that the LIR motif of BLTP3A is necessary for such recruitment.

**Figure 5. Activation of CASM recruits BLTP3A to the surface of damaged lysosomes through an interaction with mATG8 proteins.**

(A) Alignment of the region of BLTP3A orthologues from different species centered on the a.a. region required for mATG8 binding in human BLTP3A. The alignment shows a high degree of conservation of the key residues of the LC3-interacting region (LIR) motif among several chordates and also observed in flies. (B) Live fluorescence images (inverted grays) of RPE-1 cells expressing BLTP3A-mRFP (top) or GFP-LC3B (bottom) in either fed (left) or starved (right) conditions. Arrows (magenta) indicate GFP-LC3B positive foci. Scale bar, 5 μm. (C) Time-series of live fluorescence images (inverted grays) of exogenous BLTP3A-mRFP and GFP-LC3B before and after addition of LLOMe. Arrowheads point to lysosomes where BLTP3A and LC3B decorate the entire lysosome profile upon addition of LLOMe. Scale bar, 5 μm. (D) Time-series of live fluorescence images (inverted grays) of exogenous BLTP3AΔLIR-mRFP and GFP-LC3B before and after addition of LLOMe. Arrowheads point to lysosomes where LC3B, but not BLTP3A lacking a LIR motif, decorates the entire lysosome profile upon addition of LLOMe. Scale bar, 5 μm. (E) Correlative fluorescence-FIB-SEM microscopy of GFP-LC3B and BLTP3A-mRFP-positive lysosomes in an RPE-1 cell 15 min after LLOMe addition. Left: Fluorescence image of an RPE-1 cell expressing GFP-LC3B (green), BLTP3A-mRFP (magenta), and mito-BFP (not shown), 15 min after addition of LLOMe. Scale bar, 5 μm. Region of cell for FIB-SEM marked by dotted line. Middle: High magnification of the RPE-1 cell at left (blue square) showing both the BLTP3A-mRFP (magenta) and the GFP-LC3B (green) channels. Individual channels are shown as inverted grays. Merged channel also shows mito-BFP (blue) used for aligning EM and fluorescence images. Scale bar, 2 μm. Right: EM reconstruction (scale bar, 1 μm) corresponding to the blue square from left. Lysosome, dark green; ER, yellow; mitochondria, blue; small vesicles, magenta.

The absence of large vesicle clusters around LC3-positive lysosomes was further corroborated by correlative fluorescence-focused ion beam scanning electron microscopy (FIB-SEM), which allowed us to obtain views of the entire surface of lysosomes in RPE-1 cells positive for exogenous BLTP3A-mRFP and GFP-LC3B after 15 min of LLOMe exposure (Fig. 5E). Only scattered vesicles were observed on such lysosomes (Fig. 5E), in strong contrast with the massive accumulation of vesicles observed in the absence of LLOMe (Fig. 1H). Importantly, correlative fluorescence-FIB-SEM microscopy also showed abundant presence of ER contacts (Fig. 5E; Movies EV8 and 9), as expected after LLOMe treatment (Tan and Finkel, 2022; Radulovic et al, 2022; Wang et al, 2025).

## Recruitment of BLTP3A to vacuoles containing internalized crystals corroborates its identification as a CASM effector

To validate our model that BLTP3A is an effector of CASM, we used an alternative manipulation to LLOMe and GPN to activate this pathway. Monosodium urate (MSU) crystals, which are readily taken up by phagocytosis and accumulate within intracellular vacuoles, are known activators of CASM (Cross et al, 2023). MSU crystals added to RPE-1 cells for 2 h were easily observed by brightfield microscopy (Fig. 6A). After internalization they accumulated in LAMP1-positive vacuoles, many of which were decorated by LC3, consistent with these crystals being CASM activators (Cross et al, 2023). In cells expressing BLTP3A-mRFP along with LAMP1-GFP or GFP-LC3, a strong co-localization of BLTP3A-mRFP, but not of BLTP3A lacking the LIR motif

(BLTP3AΔLIR-mRFP) was observed with these proteins on crystal containing vacuoles (Fig. 6B,C). We also used time-lapse fluorescence microscopy to capture the moments of CASM activation (i.e. the accumulation of GFP-LC3B along the surface of a crystal-containing vacuole) and observed a very close temporal correlation of GFP-LC3B and BLTP3A-mRFP recruitment (Fig. 6D; Movie EV10), in agreement with BLTP3A being an effector of CASM.

## Loss of BLTP3A in A549 cells impacts lysosome homeostasis

In view of the finding that BLTP3A is a factor responding to lysosome perturbation, we explored the possibility that BLTP3A may have an impact on lysosome properties and/or in counteracting lysosome damage. To this aim we generated BLTP3A KO cells. The BLTP3A gene was edited in A549 cells to include frameshift mutations in exon 2 (Fig. EV3D) and absence of the BLTP3A protein in lysates from these cells was confirmed by western blotting (Fig. 7A). Interestingly, while levels of some proteins of the late endosomal system, for example Rab7, were unchanged in the lysates of KO cells relative to controls, a ~fivefold decrease in the protein levels of LAMP1 was observed (Figs. 7A and EV3E), confirmed by a decrease of anti-LAMP1 immunofluorescence, despite no obvious difference in lysosome number (Fig. 7B), pointing to an important role of BLTP3A in lysosome homeostasis.

In order to assess the impact of the lack of BLTP3A on lysosome damage, we used the galectin-3 (Gal3)-based immunofluorescence assay (Fig. 7C) (Jia et al, 2020). Gal3, which is normally diffuse in the cytosol, binds to β-galactosides and accumulates within lysosomes only when the membrane of these organelles is damaged. This accumulation results in bright puncta of anti-Gal3 immunofluorescence which corresponds to lysed lysosomes. Upon addition of LLOMe to A549 cells, bright puncta of Gal3 immunoreactivity started to appear in both wild-type and BLTP3A KO cells, (Fig. 7C). However, at the 30 and 60 min timepoints, a higher number of Gal3 spots were observed in BLTP3A KO cells relative to wild-type cells (Fig. 7D) revealing a greater fragility of KO cells. This effect was rescued by over-expressing wild-type BLTP3A-RFP, but not by expressing BLTP3AΔLIR-RFP, in BLTP3A KO cells, while only a small reduction was found in KO cells expressing BLTP3AΔLIR-RFP (Fig. EV4A,B). These findings suggest that the recruitment of BLTP3A to lysosomes may play a role in the resilience and/or response of these organelles to damage.

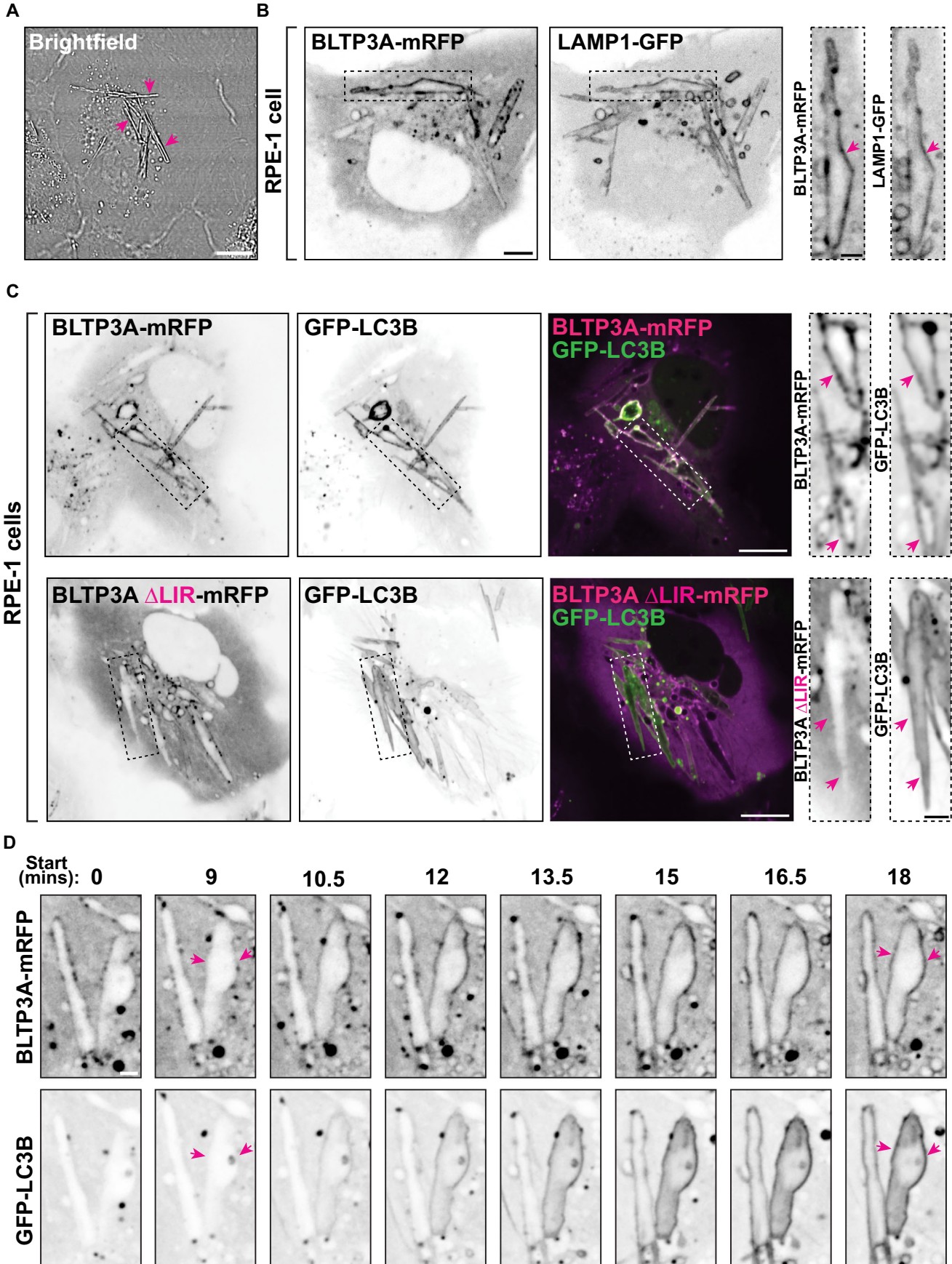

**Figure 6.  LC3-dependent BLTP3A recruitment to vacuoles containing internalized MSU crystals via CASM activation.**

(A) Brightfield image of RPE-1 cells incubated with MSU crystals (200 μg/mL) for 2 h. Arrowheads indicate MSU crystals. Scale bars, 5 μm. (B) Live fluorescence images of RPE-1 cells expressing BLTP3A-mRFP and LAMP1-GFP (shown as inverted greys) and incubated with MSU crystals for 2 h. Scale bar, 5 μm. Zoomed images (dotted rectangles) of individual channels are shown to the right. Scale bars, 2 μm. Arrowheads point to the surface of crystal-containing vacuoles positive for BLTP3A-mRFP and LAMP1-GFP. (C) Live fluorescence images of RPE-1 cells co-expressing either BLTP3A-mRFP (top row) or BLTP3AΔLIR-mRFP (bottom row) with GFP-LC3B and incubated with MSU crystals for 2 h. Individual channels are shown as inverted greys. Scale bar, 10 μm. Zoomed images (dotted rectangles) of individual channels are shown to the right. Scale bar, 2 μm. Arrowheads point to the surface of the crystal-containing vacuoles positive for GFP-LC3B to highlight the presence of BLTP3A-mRFP (top) and the absence of BLTP3AΔLIR-mRFP (bottom). (D) Time-series of live fluorescence images (inverted grays) of BLTP3A-mRFP and GFP-LC3B in RPE-1 cells incubated with MSU crystals. Scale bar, 2 μm. Arrowheads point to the surface of crystal-containing vacuoles.

# Discussion

Our study shows that BLTP3A, a protein expected to transfer lipids between adjacent membranes via a bridge-like mechanism, is a component of protein networks implicated in membrane traffic in late endosomes/lysosomes and that lysosome damage has an impact on its localization. BLTP3A binds and clusters vesicles of the endocytic system positive for VAMP7 and VAMP4 and tethers them to lysosomes via an interaction of its N-terminal region with lysosome-bound Rab7. Upon lysosome membrane damage and the subsequent recruitment to their surface of mATG8 family proteins via CASM, this Rab7-dependent interaction is disrupted. However, within minutes, BLTP3A then reassociates with lysosomes by interacting with mATG8 proteins via a LIR motif present in a long predicted unfolded loop emerging from the C-terminal region of the protein. As we discuss below, we suggest that BLTP3A may cooperate with other BLTPs in the response of cells to lysosome damage. Accordingly, we have found evidence for an increased fragility of lysosomes in response to damage in BLTP3A KO cells.

The property of BLTP3A to associate with small vesicles and induce their striking accumulation when overexpressed is shared with BLTP3B (Hanna et al, 2022). BLTP3A- and BLTP3B-positive vesicles have similar size, share at least one cargo, ATG9A, and clusters of them are closely associated to other organelles. However, these organelles differ for the two proteins: Rab5 positive early endosomes in the case of BLTP3B and Rab7-positive lysosomes in the case of BLTP3A. Moreover, BLTP3A and BLTP3B-positive vesicles differ at least partially in protein composition, as only BLTP3A-positive clusters are strongly immunolabeled by antibodies directed against VAMP7 and VAMP4.

BLTP3 vesicle clusters are reminiscent of the clusters of synaptic vesicles at synapses. Such clusters were shown to have the properties of liquid biomolecular condensates (Milovanovic and De Camilli, 2017; Milovanovic et al, 2018; Park et al, 2021). Similarly, vesicle clusters involving BLTP3A and BLTP3B are very dynamic (see for example, movies of these clusters in (Hanna et al, 2022)). The assembly of BLTP3A-positive vesicle clusters and their anchorage to endosomes/lysosomes, implies a minimum of three direct or indirect interactions of BLTP3A: 1) an interaction with vesicles, 2) an interaction with itself or with adaptor proteins to cluster vesicles and 3) an interaction with lysosomes to account for the anchoring of the vesicle clusters to these organelles.

We have shown that the interaction with lysosomes of vesicle-associated BLTP3A (interaction #3) is mediated by the binding to lysosome-bound Rab7, consistent with its being a Rab7 effector. Moreover, our results suggest that such an interaction involves the N-terminal region of BLTP3A, where we have detected the Rab7

binding site. We have also shown that the binding to vesicles (interaction #1) is mediated by the opposite end of the protein, i.e. its C-terminal region, but so far we have not identified a vesicle binding partner by co-immunoprecipitation. We note that even in the case of synaptic vesicle condensates (Milovanovic et al, 2018; Park et al, 2021), which involve interactions between synapsin (a cytosolic protein) (Südhof et al, 1989; De Camilli et al, 1990) and synaptophysin (a vesicle protein) (Johnston et al, 1989), a direct interaction between the two proteins could not be detected by co-precipitation (Park et al, 2021), although clearly such an interaction occurs in living cells where low affinity is counteracted by multivalency: a multiplicity of low affinity interactions between synapsin with itself and with synaptophysin, a protein present in multiple copies on the vesicles.

Concerning the mechanisms responsible for vesicle clustering (interaction #2), these may involve self-association of BLTP3A via low complexity unfolded sequences that project out of its rod-like core, or binding of BLTP3A to yet to be discovered adaptor/crosslinker proteins (Park et al, 2021). As the property to cluster vesicles is shared by both BLTP3A and BLTP3B, such a property must rely on shared molecular determinants of these two proteins. Ongoing work is addressing these mechanisms. The property of BLTP3 proteins to cluster vesicles into small packages may be an important aspect of their physiological function as BLTP3B-positive vesicles were shown to be organized in small clusters even at physiological levels of expression (Hanna et al, 2022). Large clusters likely result from BLTP3 overexpression, although other scenarios, such as that BLTP3 overexpression may result in vesicle accumulation due to a dominant negative effect on their fusion with downstream targets, cannot be ruled out.

Our study indicates that the Rab7-dependent association of BLTP3A with lysosomes requires a portion of BLTP3A located in proximity of its N-terminal region, the so-called chorein motif (Kumar et al, 2018). In most other BLTP family members, the chorein or chorein-like motifs are localized at the ER, which is typically thought to be the "donor" membrane in their lipid transfer function (Kumar et al, 2018; Valverde et al, 2019; Guillén-Samander et al, 2021; Cai et al, 2022; Levine, 2019; Osawa et al, 2019). This is also the case for the other two BLTPs reported to be implicated in ER to lysosomes lipid flux, VPS13C (Kumar et al, 2018; Wang et al, 2025) and ATG2 (Tan and Finkel, 2022; Cross et al, 2023). Thus, we consider it unlikely that the orientation of BLTP3A when cross-linking vesicles to Rab7 may be relevant to its lipid transfer properties.

Lysosomal membrane damage activates multiple response mechanisms at timepoints ranging from seconds to hours (Skowyra et al, 2018; Radulovic et al, 2018; Tan and Finkel, 2022; Meyer and

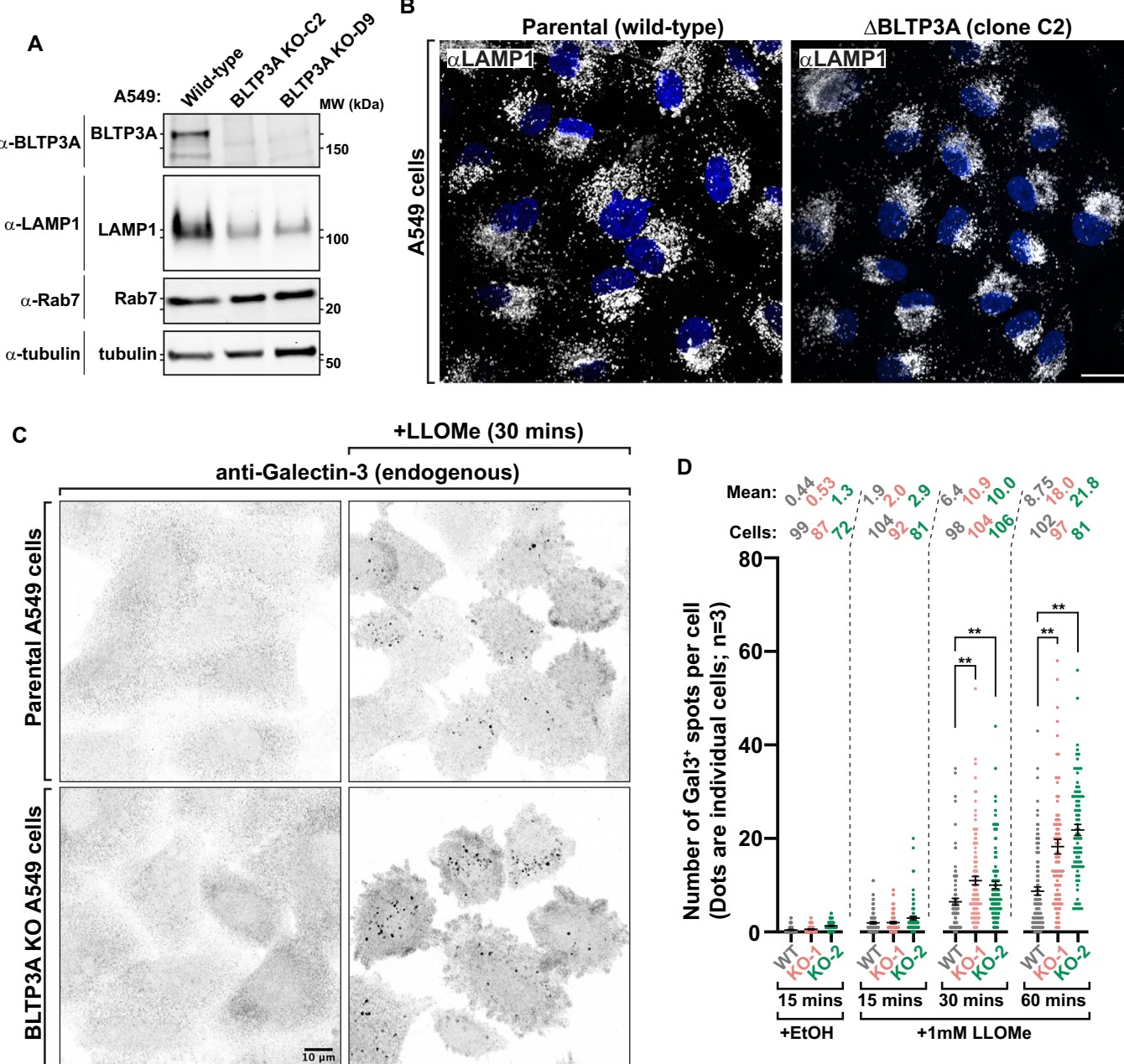

**Figure 7. Lysosomes are more susceptible to damage in cells lacking BLTP3A.**

(A) Western blots of cell extracts (inputs) of control and edited A549 cells (BLTP3A KO clones) for the proteins indicated. (B) Fluorescence images of parental control (left) or BLTP3A KO (right) A549 cells with antibodies against LAMP1 (green). Scale bar, 20 μm. (C) Fluorescence images of parental control (top row) or BLTP3A KO (bottom row) A549 cells with antibodies against galectin-3 (inverted greys). Cells were treated with vehicle control (left column) or 1 mM LLOMe (right column). Scale bar, 10 μm. (D) Quantification of galectin-3 spots per cell from field (C) (N = 3, biological replicates). Error bars report the standard error of the mean (SEM). **$P < 0.01$. Mean number of galectin-3 spots per cell and number of cells counted per condition indicated.

Kravic, 2024; Wang et al, 2025). Conjugation of mATG8 proteins to the surface of lysosomes (CASM pathway) is one such response (Durgan and Florey, 2022). mATG8 proteins can be detected on lysosomes ~5–10 min after the initiation of membrane damage, similar to the time frame of BLTP3A reassociation. It is therefore of great interest that 1) top hits identified from our immunoprecipitation of endogenously tagged BLTP3A^V5 from A549 cells are two

mATG8 proteins (MAP1LC3B and GABARAP) and 2) BLTP3A contains a LIR motif (i.e. an Atg8 binding consensus) within a disordered loop projecting out of the C-terminal rod-like region. Accordingly, we found that the reassociation of BLTP3A after LLOMe follows the accumulation of LC3B on the surface of lysosomes and that the LIR motif is required for the reassociation of BLTP3A with their surface.

The LIR motif of BLTP3A is localized on a long (305 a.a.) predicted unfolded loop emerging from the C-terminal portion of BLTP3A, which is the portion of the protein expected to bind vesicles. However, although clusters of BLTP3A-positive vesicles persist in the cytoplasm, the bulk of the pool of BLTP3A that reassociates with the lysosomal surface is not bound to small vesicles, suggesting that the LIR motif-dependent interaction of BLTP3A with lysosome-bound LC3B is mutually exclusive with its association with vesicles. A binding of BLTP3A to lysosomes via its LIR motif, rather than by Rab7, would leave its N-terminal chorein domain region, no longer engaged by Rab7, available for an interaction with the ER, a possibility that needs to be further explored.

Lysosome membrane damage is known to result in the formation of ER-lysosome tethers comprising other lipid transfer proteins, such as ORP proteins (Tan and Finkel, 2022; Radulovic et al, 2022), VPS13C (Wang et al, 2025) and ATG2 (Tan and Finkel, 2022; Cross et al, 2023), most likely as a cellular response aimed at protecting or repairing membranes by delivering new lipids. We suggest that under these conditions BLTP3A may be available on the surface of damaged lysosomes to function at close appositions between lysosomes and the ER where it may cooperate with other BLTPs in transferring membrane lipids to lysosomes.

However, we note an important limitation of this study: although the structure and biochemical properties of BLTP3A suggest that it may transfer lipids between two membranes via a bridge-like mechanism, the precise mechanism(s) through which this putative function plays a role in lysosome biology and in response to lysosome damage remains unclear and is the primary focus of our future studies. As lysosomes play a key role in cells of the immune system, it is of special interest that several coding variants of BLTP3A are associated with susceptibility to systemic lupus erythematosus (SLE) (Gateva et al, 2009; Zhang et al, 2011; Wen et al, 2020), a chronic autoimmune disease.

# Methods

### Reagents and tools table

| Reagent/resource | Reference or source | Identifier or catalog number |
|---|---|---|
| **Antibodies** | | |
| Rabbit anti-V5 | Cell Signaling Technology | Cat# 13202; RRID:AB_2687461 |
| Rabbit anti-UHRF1BP1 (BLTP3A) | Bethyl Laboratories | Cat# A304-646A; RRID:AB_2620841 |
| Mouse anti-alpha tubulin | Sigma-Aldrich | Cat# T6793, RRID:AB_477585 |
| Mouse anti-VAMP7 | Synaptic Systems | Cat# 232 011, RRID:AB_2619947 |
| Mouse anti-VAMP4 | Proteintech | Cat# 67219-1-Ig, RRID:AB_2882510 |
| Rabbit anti-ATG9A | Abcam | Cat# ab108338, RRID:AB_10863880 |
| Mouse anti-GAPDH | Thermo Fisher Scientific | Cat# MA5-15738, RRID:AB_10977387 |
| Mouse anti-vinculin | Millipore-Sigma | Cat# SAB4200729, RRID:AB_2877646 |
| Mouse anti-LAMP1 | DSHB | Cat# H4A3, RRID:AB_2296838 |
| Rat anti-Galectin 3 | BioLegend | Cat# 125402, RRID:AB_1134238 |
| Mouse anti-Rab7 | Cell Signaling Technology | Cat# 95746S, RRID:AB_2800252 |
| Rabbit anti-pRab7 | Abcam | Cat# ab302494, RRID:AB_2933985 |
| **Recombinant DNA** | | |
| pmCh-N1 BLTP3A | De Camilli lab | RRID:Addgene_241246 |
| pEGFP-N1 BLTP3A | De Camilli lab | RRID:Addgene_241247 |
| pmCh-N1 BLTP3B | De Camilli lab | RRID:Addgene_241248 |
| pmCh-N1 BLTP3A LIR mutant | De Camilli lab; this manuscript | RRID:Addgene_241249 |
| pmCh-N1 BLTP3A 1-336 | De Camilli lab; this manuscript | RRID:Addgene_241250 |
| pmCh-N1 BLTP3 chimera-1 | De Camilli lab; this manuscript | RRID:Addgene_241251 |
| pmCh-N1 BLTP3 chimera-2 | De Camilli lab; this manuscript | RRID:Addgene_241252 |
| pmCh-N1 BLTP3A 1-1327 | De Camilli lab; this manuscript | RRID:Addgene_241253 |
| pmCh-N1 BLTP3A 1-1364 | De Camilli lab; this manuscript | RRID:Addgene_241254 |
| pmCh-C1 Rab45 | De Camilli lab | RRID:Addgene_241255 |
| PX458 BLTP3A knockout | De Camilli lab; this manuscript | RRID:Addgene_241256 |
| mScarlet-LRRK1 D1409A | De Camilli lab; credit to Xinbo Wang | RRID:Addgene_241257 |
| pEGFP-N1 LAMP1 | Addgene | RRID:Addgene_34831 |
| GFP-LRRK1 K746G | MRC Reagents and Services | DU67083 |
| RFP-LRRK1 K746G | De Camilli lab | RRID:Addgene_233586 |
| GFP-Rab7 wt | Addgene | RRID:Addgene_61803 |
| GFP-Rab7 Q67L | Addgene | RRID:Addgene_169038 |
| GFP-Rab7 T22N | Addgene | RRID:Addgene_28048 |
| pmApple IST1 | Kind gift from Phyllis Hanson, University of Michigan School of Medicine, Ann Arbor, MI | An RRID does not exist for this resource. |
| NPC1-GFP | Addgene | RRID:Addgene_53521 |
| GFP-LC3B | Addgene | RRID:Addgene_11546 |
| mito-BFP | Addgene | RRID:Addgene_49151 |
| **Experimental models** | | |
| hTERT-RPE1 | ATCC | CRL-4000; RRID:CVCL_4388 |
| A549 wild-type | ATCC | CCL-185; RRID:CVCL_0023 |
| A549 BLTP3A^V5, B6 clone | De Camilli lab | RRID:CVCL_E9W3 |
| A549 BLTP3A^V5, C5 clone | De Camilli lab | RRID:CVCL_E9W4 |

| Reagent/resource | Reference or source | Identifier or catalog number |
| --- | --- | --- |
| A549 BLTP3A knockout, Clone D9 | De Camilli lab | RRID:CVCL_E9W2 |
| A549 BLTP3A knockout, Clone C2 | De Camilli lab | RRID:CVCL_E9W1 |
| A549 VPS13C knockout | De Camilli lab | RRID: CVCL_E6IP |
| C57BL/6J mice | Jackson Laboratory | RRID:IMSR_JAX:000664 |
| **Software** | | |
| ImageJ | Fiji | RRID:SCR_002285 |
| GraphPad Prism | GraphPad Prism (RRID:SCR_002798) | RRID:SCR_002798 |
| AlphaFold | AlphaFold (RRID:SCR_025454) | RRID:SCR_025885 |
| ConSurf Database | ConSurf Database (RRID:SCR_002320) | RRID:SCR_002320 |
| Velox | Thermo FIsher Scientific | An RRID does not exist for this resource. |
| SIFT based MATLAB script | Github/Zenodo | https://doi.org/10.5281/zenodo.15677643 |
| ImageJ code: Analysis script_0 | Github/Zenodo | https://doi.org/10.5281/zenodo.15653077 |
| Python code: Figure generating script | Github/Zenodo | https://doi.org/10.5281/zenodo.15652924 |
| **Chemicals, enzymes and other reagents** | | |
| FuGene HD | Promega | Cat# E2311 |
| Lipofectamine 2000 | Invitrogen | Cat# 11668019 |
| PolyJet™ | SignaGen Laboratories | Cat# SL100688 |
| Opti-Mem | Thermo Fisher Scientific | Cat# 31985062 |
| SDS | American Bio | Cat# AB01920-00500 |
| Tris | American Bio | Cat# AB02000-05000 |
| NaCl | Sigma-Aldrich | Cat# 3624-05 |
| Tween-20 | Sigma-Aldrich | Cat# P7949 |
| Bromphenol Blue | Sigma-Aldrich | Cat# B5525 |
| B-mercaptoethanol | Sigma-Aldrich | Cat# M3148 |
| Glycerol | American Bio | Cat# AB00751 |
| Methanol | Sigma-Aldrich | Cat# 179337-4L-PB |
| COmplete mini EDTA Free | Roche | Cat# 11836170001 |
| Precision Plus Protein Dual Color Standards | Biorad | Cat# 1610374 |
| 4–15% MiniPROTEAN 10-well | Biorad | Cat# 4568084 g |
| 4–15% MiniPROTEAN 15-well | Biorad | Cat# 4568086 |
| Nitrocellulose Membrane, Precut, 0.2 μm, 7 ×8.4 cm | Biorad | Cat# 1620146 |
| 10x Tris/Glycine Buffer | Biorad | Cat# 1610771 |

| Reagent/resource | Reference or source | Identifier or catalog number |
| --- | --- | --- |
| 10x Tris/Glycine/ SDS Buffer | Biorad | Cat# 1610772 |
| Nonfat Dry Powder Milk | Research Products International | Cat# M17200-500.0 |
| Odyssey Classic Imager | LICORbio | RRID:SCR_023765 |
| Dragonfly 200 Confocal Microscope | Andor | An RRID does not exist for this resource. |
| BSA | Sigma-Aldrich | Cat# A9647 |
| Saponin | Sigma-Aldrich | Cat# 84510, CAS: 8047-15-2 |
| Paraformaldehyde 16% Aqueous Solution EM Grade | Electron Microscopy Sciences | Cat# 15710 |
| ProLong™ Gold Antifade Mountant with DNA Stain DAPI | Thermo Fisher Scientific | Cat# P36935 |
| Fisherbrand™ Superfrost™ Disposable Microscope Slides | Thermo Fisher Scientific | Cat# 12-550-143 |
| Circular Cover Glass, 18 mm, #1½, 1 oz | Electron Microscopy Sciences | Cat# 72222-01 |
| Ampicillin sodium salt | Sigma-Aldrich | Cat# A0166, CAS: 69-52-3 |
| Kanamycin | Sigma-Aldrich | Cat# K1377, CAS: 25389-94-0 |
| LB Broth | BD | Cat# 244610 |
| HIFI DNA Assembly Master Mix | NEB | Cat# E2621L |
| NEB® 5-alpha Competent E. coli (High Efficiency) | NEB | Cat# C2987H |
| One Shot™ TOP10 Chemically Competent E. coli | Thermo Fisher Scientific | Cat# C404003 |
| Ethanol | Decon Laboratories | Cat# 2716 |
| GoTaq® DNA Polymerase | Promega | Cat# M3001 |
| Phusion™ Hot Start II DNA Polymerase | Thermo Fisher Scientific | Cat# F549L |
| Bacteriological Petri Dishes | Fisher | Cat# 08757100D |
| Leu-Leu methyl ester hydrobromide (LLOMe) | Sigma-Aldrich | Cat# L7393-500MG, CAS: 1668914-8 |
| GPN | Santa Cruz Biotechnology | Cat# sc-252858; CAS: 14634 |
| Monosodium Urate Crystals | AdipoGen | Cat# AGCR139502002 |
| Alt-R™ CRISPR-Cas9 tracrRNA, ATTO™ 550, 5 nmol | Integrated DNA Technologies | Cat# 1075934 |
| Alt-R™ S.p. HiFi Cas9 Nuclease V3, 100 μg | Integrated DNA Technologies | Cat# 1081060 |

| Reagent/resource | Reference or source | Identifier or catalog number |
|---|---|---|
| Alt-R™ HDR Enhancer V2, 30 µL | Integrated DNA Technologies | Cat# 10007910 |
| Geneticin™ Selective Antibiotic (G418 Sulfate) (50 mg/mL) | Thermo Fisher Scientific | Cat# 10131027 |
| V5-Trap® Magnetic Agarose | ChromoTek | Cat# v5tma-20 |
| Triton X-100 | American Bioanalytical | Cat# ab02025 |
| Protease inhibitor tablets | Sigma-Aldrich | Cat# 11836170001 |
| FEI Talos L120C TEM and STEM | Thermo Fisher Scientific | RRID:SCR_019908 |
| Zeiss Versa XRM-620 | Zeiss | An RRID does not exist for this resource. |
| Leica EM ACE600 High Vacuum Sputter Coater | Leica Microsystems | RRID:SCR_020237 |
| 35 mm Dish No. 1.5 Gridded Coverslip 14 mm Glass Diameter | MatTek | Cat# P35G-1.5-14-CGRD |
| Aqueous Glutaraldehyde EM Grade 25% | Electron Microscopy Sciences | Cat# 16220 |
| Sodium Cacodylate Buffer, 0.2 M, pH 7.4 | Electron Microscopy Sciences | Cat# 11652 |
| Osmium Tetroxide 4% Aqueous Solution | Electron Microscopy Sciences | Cat# 19140 |
| Potassium hexacyanoferrate(II) trihydrate, $K_4Fe(CN)_6$ | Sigma-Aldrich | Cat# P3289, CAS: 14459-95-1 |
| Uranyl Acetate | Electron Microscopy Sciences | Cat# 22400, CAS: 541-09-3 |
| EMbed 812 Resin | Electron Microscopy Sciences | Cat# 14900 |
| UranyLess EM Stain | Electron Microscopy Sciences | Cat# 22409 |
| Lead Citrate 3% | Electron Microscopy Sciences | Cat# 22410 |
| Durcupan™ Acm Epoxy Resin | Electron Microscopy Sciences | Cat# 14040 |
| DL-dithiothreitol (DTT) | AmericanBio | Cat# AB00490-00010 |
| Ammonium bicarbonate (ABC) | Thermo Scientific | Cat# 393212500 |
| Urea | Thermo Scientific | Cat# 434720050 |
| 2-iodoacetamide (IAA) | Sigma-Aldrich | Cat# I6125-5G |
| trifluoroacetic acid (TFA) | Thermo Scientific | Cat# 85183 |
| acetonitrile (ACN) | Sigma-Aldrich | Cat# 900667-100 ML |
| Formic acid (FA) | Thermo Fisher Scientific | Cat# 28905 |
| HPLC-grade water | Thermo Fisher Scientific | Cat# 268300010 |

| Reagent/resource | Reference or source | Identifier or catalog number |
|---|---|---|
| Trypsin Protease | Thermo Fisher Scientific | Cat# 90058 |
| HEPES | Sigma-Aldrich | Cat# H4034-100G |
| Live Cell Imaging Solution | Invitrogen | Cat# A59688DJ |

## Antibodies and reagents

The list of plasmids, antibodies, their working dilution, and the supplier for this study can be found in the Dataset EV1 (Key Resource Table).

## Generation of plasmids

All BLTP3A and BLTP3B ORFs used in this study utilized a human codon optimized sequence designed and purchased from Genscript. Codon optimized human BLTP3 chimeras were amplified using PCR from the pcDNA3.1 plasmid and ligated into a pmCh-N1 plasmid. Most constructs were generated with regular cloning protocols or through site-directed mutagenesis. The desired ORFs were amplified by PCR and inserted into plasmids through enzyme digestions and ligation. Some amplified ORFs were ligated using HiFi assembly (NEB). Details of primer sets, enzymes, techniques, and plasmids used for each construct can be found in Dataset EV1 and in Dataset EV2.

A detailed protocol for the molecular cloning of BLTP3 plasmids for expression in mammalian cells is at: https://doi.org/10.17504/protocols.io.8epv5z5kjv1b/v1.

## Correlative light and electron microscopy

For TEM CLEM, RPE-1 cells were plated on 35-mm gridded, glass-bottom MatTek dish (P35G-1.5-14-CGRD) and transfected as described above with BLTP3A-mRFP, LAMP1-GFP, mito-BFP. Cells were pre-fixed in 4% PFA in dPBS then washed with dPBS before fluorescence light microscopy imaging. Regions of interest were selected and their coordinates on the dish were identified using phase contrast. Cells were further fixed with 2.5% glutaraldehyde in 0.1 M sodium cacodylate buffer, postfixed in 2% $OsO_4$ and 1.5% $K_4Fe(CN)_6$ (Sigma-Aldrich) in 0.1 M sodium cacodylate buffer, en bloc stained with 2% aqueous uranyl acetate, dehydrated in graded series of ethanols (50%, 75%, and 100%), and embedded in Embed 812. Cells of interest were relocated based on the pre-recorded coordinates. Ultrathin sections (50–60 nm) were post-stained with uranyl acetate substitute (UranyLess, EMS), followed by a lead citrate solution. Sections were observed in a Talos L 120 C TEM microscope at 80 kV, images were taken with Velox software and a 4k × 4 K Ceta CMOS Camera (Thermo Fisher Scientific). Except noted all reagents were from EMS (Electron Microscopy Sciences), Hatfield, PA.

A detailed protocol for 2D TEM CLEM can be found at: https://doi.org/10.17504/protocols.io.261gend2jg47/v1.

## FIB-SEM sample preparation and imaging

One Epon-embedded sample of RPE-1 cells expressing BLTP3A-mRFP and GFP-LC3B after 15 min of LLOMe exposure, was

mounted onto the top of a 1 mm copper stud using Durcupan, ensuring optimal charge dissipation by maintaining contact between the heavy metal-stained sample and the copper stud. The vertical sample post was then trimmed to a small block containing the Region of Interest (ROI), with dimensions of 80 μm in width (perpendicular to the ion beam) and 75 μm in depth (in the ion beam direction). Precise ROI targeting and trimming were achieved using overlay images from light fluorescence microscopy and X-ray tomography data acquired with a Zeiss Versa XRM-620. The detail approach was previously described by (Pang and Xu, 2023). To enhance conductivity, thin layers of conductive material—10 nm of gold followed by 40 nm of carbon—were deposited onto the trimmed sample using a Leica EM ACE600 coater.

The FIB-SEM prepared sample was imaged using a customized enhanced FIB-SEM microscope (Xu et al, 2017, 2021). The images were acquired using a 500 pA current SEM probe at 0.7 keV. The scan rate was 400 kHz, with a 2-nm pixel along $x$ and $y$ axes. A 2-nm z-step was achieved by ~5 s of milling with a 15-nA Ga$^+$ beam at 30 kV. A total volume of $12 \times 6 \times 12$ μm$^3$ was acquired over 8 days at a rate of 1 min per frame. The raw image stack was aligned using a SIFT based MATLAB script and binned 2-to-1 along $x$, $y$, and $z$ axes to create a final dataset with $4 \times 4 \times 4$ nm$^3$ voxels, which can be viewed in any arbitrary orientations.

A detailed protocol for FIB-SEM can be found at: https://doi.org/10.17504/protocols.io.kqdg3wj7ev25/v1.

## Cell culture and transfections

hTERT-RPE-1 cells were a kind gift of A. Audhya (University of Wisconsin, Madison, WI). A549 and COS-7 cells were obtained from ATCC. All mammalian cells were maintained at 37 °C in humidified atmosphere at 5% CO$_2$ unless noted otherwise. A549 and COS-7 cells were grown in DMEM and RPE-1 cells in DMEM/F12 medium (Thermo Fisher Scientific) supplemented with 10% FBS, 100 U/mL penicillin, 100 mg/mL streptomycin. In all, 2 mM glutamax (Thermo Fisher Scientific) was added to all media for RPE-1 cells. All cell lines were routinely tested and always resulted free from mycoplasma contamination.

Transient transfections were carried out on cells that were seeded at least 8 h prior. All transfections of plasmids used FuGENEHD (Promega) to manufacturer's specifications for 16–24 h in complete media without antibiotics.

A detailed protocol for cell culture, transfection, immunocytochemistry, and imaging: https://doi.org/10.17504/protocols.io.eq2lyp55mlx9/v1.

## Immunoblotting and imaging procedure

All cell samples analyzed by immunoblotting were scraped from plates and harvested by centrifugation ($500 \times g$ for 5 min). The pellet was washed with ice-cold dPBS and centrifuged again in a 1.7 mL Eppendorf tube. The cell pellet was resuspended in Lysis buffer (20 mM Tris-HCl pH 7.5, 150 mM NaCl, 1% SDS, 1 mM EDTA) containing protease inhibitor cocktail (Roche). The lysate was clarified by centrifugation ($17,000 \times g$ for 10 min) and a small portion of lysate was reserved for quantification of protein concentration by Bradford. The remaining lysate was then mixed with 5× SDS sample buffer (Cold Spring Harbor) to 1× concentration and then heated to 95 °C for 3 min. Protein

samples (15–25 ug) were separated by electrophoresis on a 4–20% Mini-PROTEAN TGX gel and then subjected to standard western blot transfer and procedures. Blots were imaged using the Odyssey imaging system (LI-COR) using manufacturer's protocols. All primary antibodies used in this study are listed in Dataset EV1.

A detailed protocol for immunoblotting can be accessed at: https://doi.org/10.17504/protocols.io.bp2l6be9zgqe/v1.

## Mouse tissue lysate preparation

Tissues were collected from sacrificed WT C57BL/6J mice (Jackson Laboratory strain #000664). For each 1 g of material, 10 ml of buffer (25 mM HEPES, pH 7.4, 200 mM NaCl, 5% glycerol, protease inhibitors) was added. Mechanical lysis was performed using a glass dounce-homogenizer (15 strokes). Triton X-100 was added to 1%, and material was rotated at 4 °C for 30 min. Material was centrifuged at $1000 \times g$ to remove cell debris and the collected supernatant was centrifuged at $27,000 \times g$ for 20 min. The resulting supernatant mixed was flash frozen and stored at −80 °C until use.

A detailed protocol for the preparation of mouse tissue lysate can be accessed at: https://doi.org/10.17504/protocols.io.14egnyd3zv5d/v1.

## Live cell imaging and immunofluorescence

For all live cell microscopy cells were seeded on glass-bottom mattek dishes (MATtek corporation) 5500/cm$^2$ in complete media. Transfections were carried out as described above. Spinning-disk confocal imaging was preformed 16–24 h post transfection using an Andor Dragonfly 200 (Oxford Instruments) inverted microscope equipped with a Zyla cMOS 5.5 camera and controlled by Fusion (Oxford Instruments) software. Laser lines used: DAPI, 440 nm; GFP, 488; RFP, 561; Cy5, 647. Images were acquired with a PlanApo objective (60 × 1.45-NA). During imaging, cells were maintained in Live Cell Imaging buffer (Life Technologies) in a cage incubator (Okolab) with humidified atmosphere at 37 °C. LLOMe (Sigma-Aldrich, CAS: 1668914-8) and GPN (Cayman Chemical, CAS: 14634) were dissolved in ethanol and used at a final concentration of 1 mM for all imaging experiments. All live imaging experiments are representative of at least ten independent repeats.

Immunofluorescent experiments were performed with cells grown on #1.5 glass cover slips. Cells were fixed with 4% PFA in PBS (Gibco, 14190144) for 15 min at room temperature, washed 3× with PBS, permeabilized using antibody dilution buffer (1× PBS containing 0.2% saponin and 2% BSA) at 4 °C overnight with the indicated primary antibodies. Slides were washed three times with PBS containing 0.02% saponin to remove excess primary antibody and subsequently incubated with secondary antibodies diluted in antibody dilution buffer for 45 min at room temperature in the dark. Slides were washed again three times to remove secondary antibody with PBS containing 0.02% saponin prior to mounting.

## Lysosomotropic drug treatments

Cells (RPE-1 or A549) were seeded 16–24 h before treatments. A fresh 333 mM stock solution of LLOMe in ethanol was prepared and kept on ice. For live-imaging experiments, a 2× stock of LLOMe or GPN (2 mM) was prepared in complete media immediately before adding to an equal volume of complete media

on a Mattek dish. Cells were imaged at 37 °C with 5% $CO_2$ at all times. For lysosome damage experiments (galectin-3 immuno-fluorescence), LLOMe (1 mM) was added to complete medium pre-warmed to 37 °C and immediately added to cells. Cells were then incubated at 37 °C with 5% $CO_2$ for 15, 30, or 60 min. Untreated cells received an identical volume of vehicle (ethanol) in complete medium for 15 min and served as a baseline. After incubation with LLOMe or vehicle, cells were fixed as described above before being incubated overnight at 4 °C with antibodies against galectin-3.

## Quantification of Gal3 puncta in A549 cells

Fields of view for all Gal3 experiments were selected based on either DAPI staining to maximize the number of cells per field or by exogenous RFP signal to ensure imaged cells were expressing the desired construct for rescue experiments. Cell boundaries were drawn by hand in Fiji. Minimum display values between 2000 and 3000 and maximum display values of 5000–8000 were used to detect fluorescence. Remaining background signal after setting these minimum and maximum display values was smoothened using the "Mean" filter in Fiji set to a radius of 2 pixels. After smoothening, Fiji's default thresholding algorithm was used to identify the foreground objects and create a mask of Gal3 spots. The resulting mask was processed using Fiji's binary "Watershed" separation technique to separate foreground objects (Gal3 spots) that were touching. Finally, using Fiji's "Analyze particles" function, ROIs were generated for all mask objects and counted as individual Gal3 spots. ROIs smaller than 3 pixels were considered noise and were not included in the count.

The macro code for Gal3 quantification can be accessed at: https://doi.org/10.5281/zenodo.15653077.

## Generation of CRISPR edited cell lines

Endogenous tagging of A549 cells was carried out using a CRISPR/Cas9 ribonucleoprotein (RNP) complex. Commercially purified Cas9 nuclease (IDT) was combined and incubated with trans-activating CRISPR RNA (tracrRNA) and a BLTP3A-targeting guide RNA to form the RNP complex. The RNP complex was then incubated with a single-stranded repair template containing the insert sequence and ultimately delivered to A549 cells via electroporation. Subsequently, the RNP-transfected pools were serially diluted and plated at 1 cell per well on a glass bottom 96-well plate. Wells were expanded and tested for editing via PCR until an appropriate number of edited clones were obtained. All gRNAs used in this study are listed in Dataset EV2.

A detailed protocol for the generation of CRISPR edited cell lines can be accessed at: https://doi.org/10.17504/protocols.io.j8nlkrxo5v5r/v1.

## Immunoprecipitation of endogenously tagged BLTP3A

Wild-type or endogenously edited A549 cells expressing BLTP3A^V5 were grown to 90% confluency on 15-cm plates (Falcon). Two 15-cm plates were used for each independent replicate of wild-type and edited cells ($n = 3$). Cells were washed with dPBS (Gibco) and then scrapped from the surface of the plates into 2 mL of dPBS per plate. Cells were pelleted at $1000 \times g$ for 5 min and the dPBS supernatant was removed. Cells were resuspended in 2 mL of ice-cold lysis buffer (50 mM Tris pH 7.6,

150 mM NaCl, 1% Triton X-100, protease inhibitors [Roche]) and rotated at 4 °C for 20 min. Cell lysates were spun at $17,000 \times g$ for 10 min to clear insoluble material. The supernatants were then added to 75 µL (slurry) of anti-V5 magnetic resin (ChromoTek) and left to rotate at 4 °C for 2 h. Resin was washed twice for 5 min rotating with ice cold lysis buffer and then washed a third time without detergent for 5 min (50 mM Tris pH 7.6, 150 mM NaCl, protease inhibitors). Supernatant was removed and resin was stored at −80 °C.

A detailed protocol for the immunoisolation of endogenously tagged BLTP3A can be accessed at: https://doi.org/10.17504/protocols.io.5jyl8qyp9l2w/v1.

## Protein digestion

The beads were resuspended in 80 µL of 2 M Urea, 50 mM ammonium bicarbonate (ABC) and treated with DL-dithiothreitol (DTT) (final concentration 1 mM) for 30 min at 37 °C with shaking at 1100 rpm on a Thermomixer (Thermo Fisher). Free cysteine residues were alkylated with 2-iodoacetamide (IAA) (final concentration 3.67 mM) for 45 min at 25 °C with shaking at 1100 rpm in the dark. The reaction was quenched using DTT (final concentration 3.67 mM), and LysC (750 ng) was added, followed by incubation for 1 h at 37 °C at 1150 rpm. Finally, trypsin (750 ng) was added, and the mixture was incubated for 16 h at 37 °C with shaking at 1150 rpm.

After incubation, an additional 500 ng of trypsin was added to the sample, followed by a 2-h incubation at 37 °C at 1150 rpm. The digest was then acidified to pH <3 by adding 50% trifluoroacetic acid (TFA), and the peptides were desalted on C18 stage tips (Empore C18 extraction disks). Briefly, the stage tips were conditioned with sequential additions of: (i) 100 mL methanol), (ii) 100 µL 70% acetonitrile (ACN)/0.1% TFA, (iii) 100 mL 0.1% TFA twice. After conditioning, the acidified peptide digest was loaded onto the stage tip, and the stationary phase was washed with 100 µL 0.1% formic acid (FA) twice. Finally, the peptides were eluted with 50 µL 70% ACN/0.1% FA twice. Eluted peptides were dried under vacuum in a Speed-Vac centrifuge, reconstituted in 12 µL of 0.1% FA, sonicated and transferred to an autosampler vial. Peptide yield was quantified using a NanoDrop (Thermo Fisher).

## Mass spectrometry analyses

Peptides were separated on a 25 cm column with a 75 µm diameter and 1.7 µm particle size, composed of C18 stationary phase (IonOptics Aurora 3 1801220) using a gradient from 2% to 35% Buffer B over 90 min, followed by an increase to 95% Buffer B for 7 min (Buffer A: 0.1% FA in HPLC-grade water; Buffer B: 99.9% ACN, 0.1% FA) with a flow rate of 300 nL/min on a NanoElute2 system (Bruker).

MS data were acquired on a TimsTOF HT (Bruker) with a Captive Spray source (Bruker) using a data-independent acquisition PASEF method (dia-PASEF). The mass range was set from 100 to 1700 $m/z$, and the ion mobility range from 0.60 V.s/cm$^2$ (collision energy 20 eV) to 1.6 V.s/cm$^2$ (collision energy 59 eV), a ramp time of 100 ms, and an accumulation time of 100 ms. The dia-PASEF settings included a mass range of 400.0–1201.0 Da, mobility range 0.60–1.60, and an estimated cycle time of 1.80 s. The dia-PASEF windows were set with a mass width of 26.00 Da, mass overlap 1.00 Da, and 32 mass steps per cycle.

## DIA data analysis

Raw data files were processed using Spectronaut version 17.4 (Biognosys) and searched with the PULSAR search engine against the Homo sapiens UniProt protein database (226,953 entries, downloaded on 2022/09/23). Cysteine carbamidomethylation was set as fixed modifications, while methionine oxidation, acetylation of the protein N terminus, and deamidation (NQ) were defined as variable modifications. A maximum of two trypsin missed cleavages was allowed. Searches used a reversed sequence decoy strategy to control the peptide false discovery rate (FDR), with a 1% FDR threshold for identification. An unpaired *t* test was used to calculate *P* values for differential analysis, and volcano plots were generated based on log2 fold change and *q* value (multiple testing corrected *P* value). A *q*-value of ≤0.05 was considered the statistically significant cut-off.

A detailed protocol for protein digestion and mass spec analysis can be found at: https://doi.org/10.17504/protocols.io.x54v9o67qv3e/v1.

The mass spectrometry proteomics data have been deposited to the ProteomeXchange Consortium via the PRIDE partner repository with the dataset identifier PXD056338.

## Image processing, analysis, and statistics

Fluorescence images presented in this study are representative of cells imaged in at least three independent experiments and were processed with ImageJ software. The dimensions of some of the magnification insets or panels were enlarged using the *Scale* function on ImageJ.

Statistical analysis was performed with GraphPad Prism 10 software. Groups were compared using a two-tailed unpaired Student *t* test and results were deemed significant when a *P* value was smaller than 0.05.

## Data availability

The data, code, protocols, and key lab materials used and generated in this study are listed in a Key Resource Table alongside their persistent identifiers through Dataset EV1 and Zenodo (https://doi.org/10.5281/zenodo.15587012) and BioImage Archive (https://doi.org/10.6019/S-BIAD2092). An earlier version of this manuscript was posted to bioRxiv on 2024-09-28 (https://doi.org/10.1101/2024.09.28.615015). License: CC-BY 4.0.

The source data of this paper are collected in the following database record: biostudies:S-SCDT-10_1038-S44318-025-00543-9.

## Peer review information

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

## Acknowledgements

We thank Thomas Melia and Shawn Ferguson (Yale) for discussion and advice, James Liu (Janelia Res Labs) for advice on endogenous tagging, Dario Alessi (Dundee, UK) for the personal communication of unpublished data and Chase Amos, Hanieh Falahati, and Berak Ugur (De Camilli lab) for critical reading of the manuscript. We also thank the support provided by the FIB-SEM Collaboration Core at Yale School of Medicine. National Institutes of Health grant DA018343 (PDC); National Institutes of Health grant NS36251 (PDC); Aligning Science Across Parkinson's grant through the Michael J. Fox Foundation for Parkinson's Research ASAP-000580 (PDC); National Institutes of Health P30 CA008748 (MM).

## Author contributions

**Michael G Hanna**: Conceptualization; Resources; Data curation; Software; Formal analysis; Supervision; Validation; Investigation; Visualization; Methodology; Writing—original draft; Writing—review and editing. **Hely O Rodriguez Cruz**: Conceptualization; Resources; Data curation; Software; Formal analysis; Investigation; Methodology; Writing—review and editing. **Kenshiro Fujise**: Formal analysis; Investigation; Methodology. **Yumei Wu**: Formal analysis; Investigation; Methodology. **C Shan Xu**: Conceptualization; Software; Formal analysis; Investigation; Methodology; Writing—review and editing. **Song Pang**: Conceptualization; Software; Formal analysis; Investigation; Methodology. Writing—review and editing. **Zhuonging Li**: Investigation; Methodology. **Mara Monetti**: Funding acquisition; Investigation; Methodology. **Pietro De Camilli**: Conceptualization; Resources; Supervision; Funding acquisition; Writing—original draft; Writing—review and editing.

Source data underlying figure panels in this paper may have individual authorship assigned. Where available, figure panel/source data authorship is listed in the following database record: biostudies:S-SCDT-10_1038-S44318-025-00543-9.

## Disclosure and competing interests statement

PDC is a member of the Scientific Advisory Board of Casma Therapeutics. The authors declare no competing interests.

# Expanded View Figures

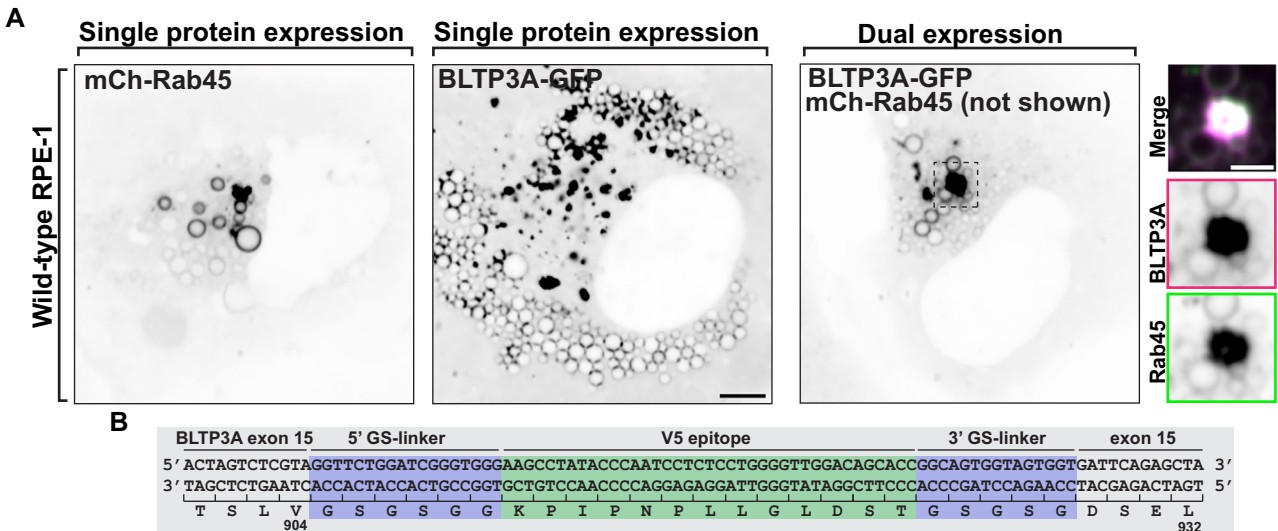

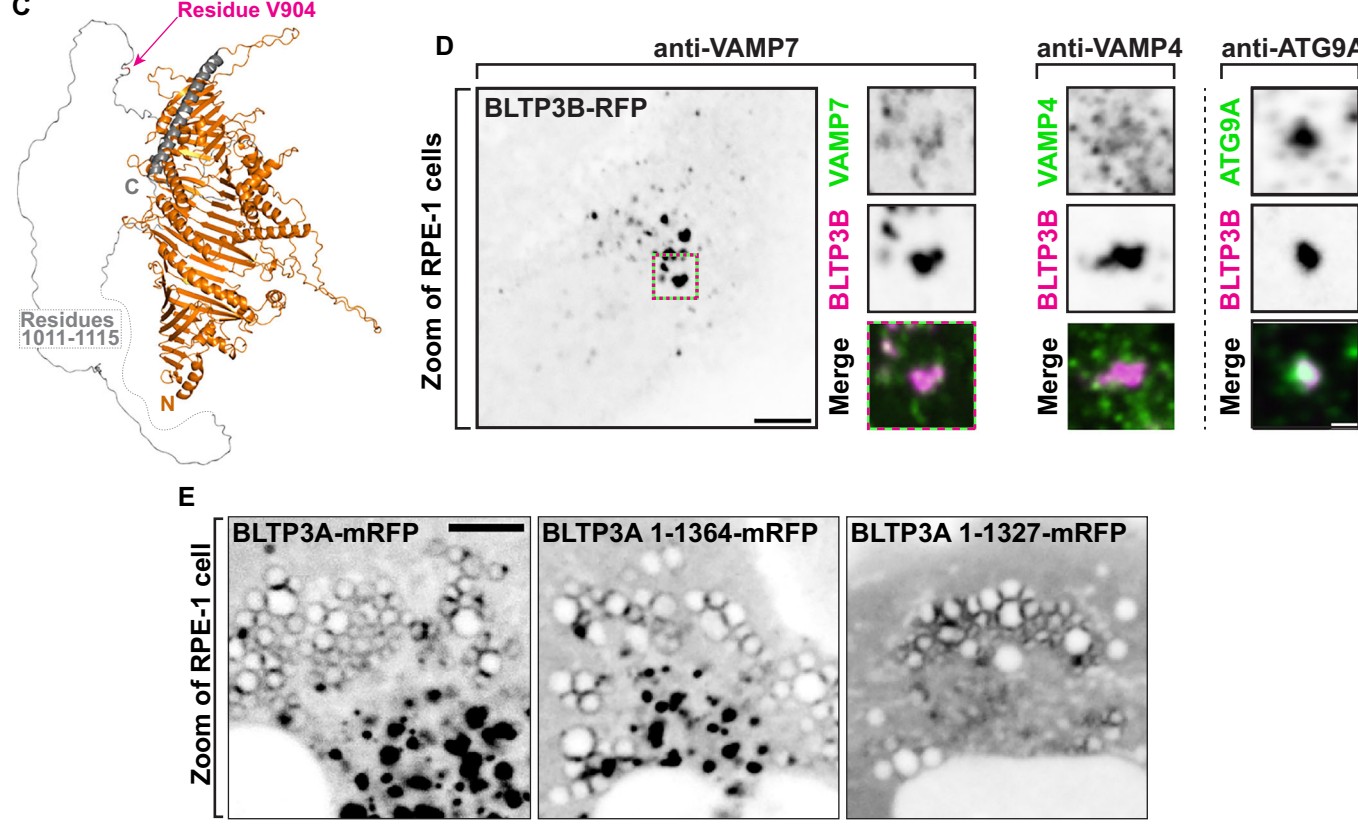

**Figure EV1. BLTP3A localizes with endo-lysosomal proteins.**

(A) Live fluorescence images (inverted grays) of RPE-1 cells expressing either GFP-Rab45 (left), BLTP3A-mRFP (center), or both proteins together (only BLTP3A is shown) (right) as indicated. Scale bar, 5 µm. High-magnification scale bar, 2 µm. (B) Genomic sequence of the edited BLTP3A locus (insertion of the V5 epitope) in A549 cell. Blue, small Gly-Ser linkers; green, V5 epitope sequence. (C) AlphaFold prediction of BLTP3A. The site where the V5 epitope (V904) was inserted is indicated. The long disordered sequence and the C-terminal helix are shown in gray. (D) Left: Fluorescence image of an RPE-1 cell expressing exogenous BLTP3B-mRFP (inverted grays) and immunolabeled with antibodies against endogenous VAMP7 (shown at right in the high magnification of the squared region in the main field). Scale bar, 5 µm. Right: zooms of different RPE-1 cells expressing exogenous BLTP3B-mRFP (magenta) and immunolabeled with antibodies (green) against endogenous VAMP4 or ATG9A. Individual channels are shown as inverted grays. Merge of channels on bottom. Scale bar, 1 µm. (E) Fluorescence images of RPE-1 cells expressing the indicated BLTP3A-mRFP construct. Scale bar, 5 µm.

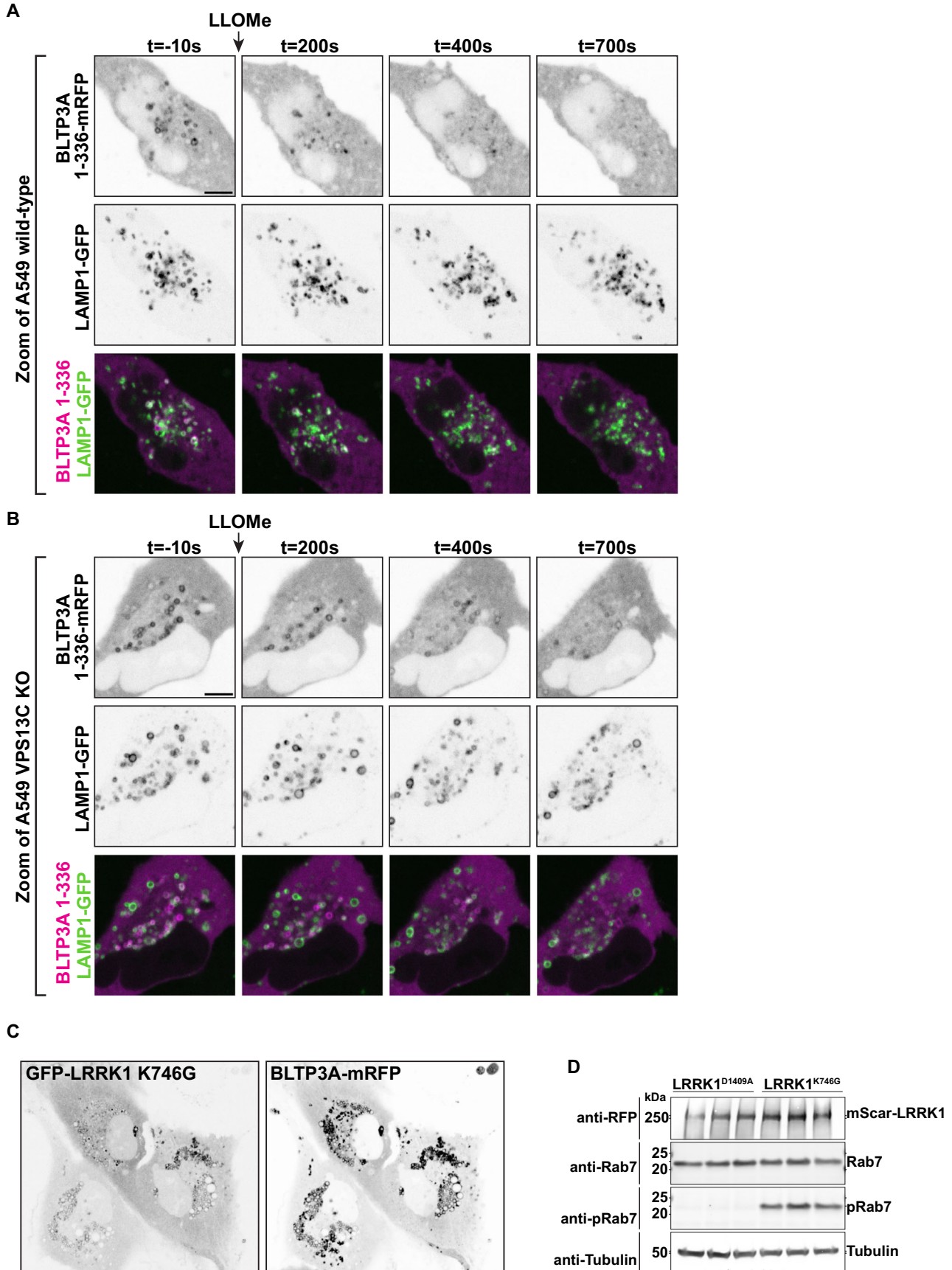

**Figure EV2.   BLTP3A is displaced from the surface of lysosomes upon lysosomal damage.**

(A) Time-series of live fluorescence images of BLTP3A-1-336-mRFP (magenta) and the lysosomal marker LAMP1-GFP (green) expressed in wild-type A549 cells before and after addition of LLOMe. Fluorescence of individual channels is shown in inverted grays. Scale bar, 5 μm. (B) Time-series of live fluorescence images of BLTP3A-1-336-mRFP (magenta) and the lysosomal marker LAMP1-GFP (green) expressed in VPS13C KO A549 cells before and after addition of LLOMe. Fluorescence of individual channels is shown in inverted grays. Scale bar, 5 μm. (C) Live fluorescence images (inverted grays) of RPE-1 cells expressing exogenous GFP-LRRK1[K746G] (left) and BLTP3A-mRFP (right). A partial association of BLTP3A-mRFP and GFP-LRRK1[K746G] was observed. Scale bar, 10 μm. (D) Western blot of lysate of RPE-1 cells expressing exogenous RFP-LRRK1[K746G] or RFP-LRRK1[D1409A] for RFP (to detect LRRK1 fusions), Rab7, phospho-Rab7 S72, and alpha-tubulin as a loading control. Individual lanes are biological replicates.

**A**

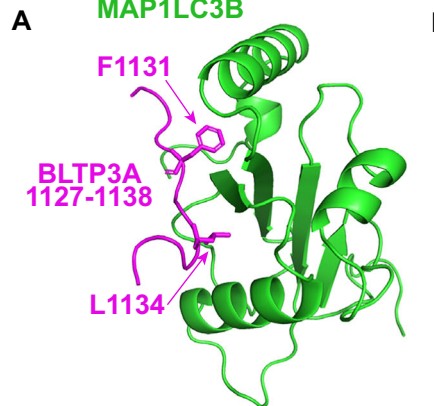

**B**

AF3 predictions of BLTP3A with mAtg8 proteins

| BLTP3A: | 1110-1150 | | 1110-1150 ΔLIR | |
|---|---|---|---|---|
| | ipTM | pTM | ipTM | pTM |
| MAP1LC3A | 0.66 | 0.7 | 0.31 | 0.69 |
| MAP1LC3B | 0.56 | 0.68 | 0.32 | 0.7 |
| MAP1LC3C | 0.57 | 0.62 | 0.32 | 0.62 |
| GABARAP | 0.74 | 0.76 | 0.55 | 0.74 |
| GABARAPL1 | 0.73 | 0.75 | 0.52 | 0.72 |
| GABARAPL2 | 0.72 | 0.74 | 0.53 | 0.73 |

**C**

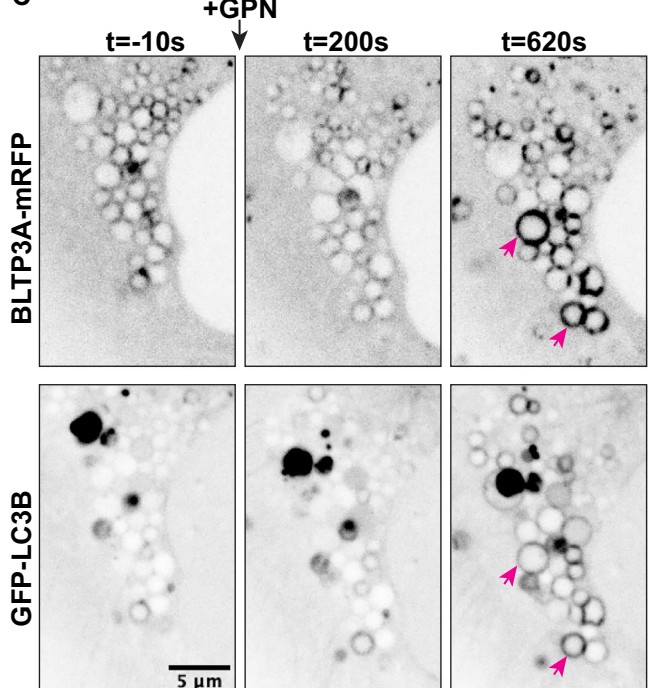

**D**

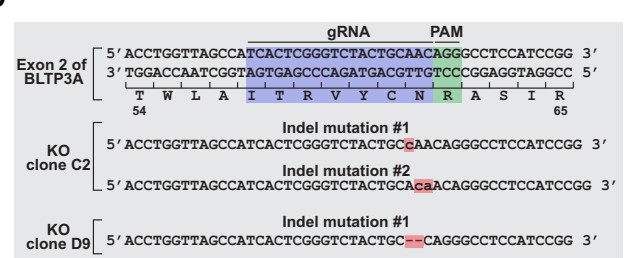

**E**

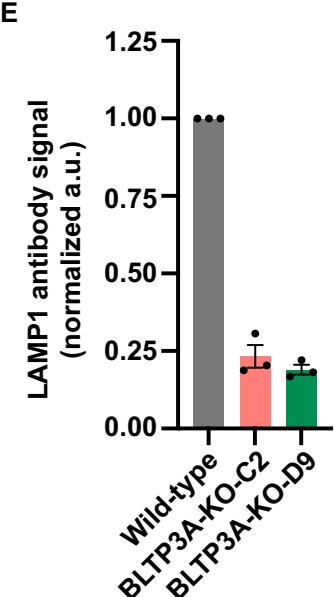

**Figure EV3.  BLTP3A is an effector of CASM.**

(**A**) AlphaFold3 multimer prediction of full-length MAP1LC3B (green) and a.a. 1110–1150 of BLTP3A (magenta). Arrows indicate key residues of the LIR motif of BLTP3A. (**B**) AlphaFold3 multimer predictions of mATG8 proteins and a.a. 1110–1150 of BLTP3A with and without the LIR motif (ΔLIR). (**C**) Time-series of live fluorescence images (inverted grays) of BLTP3A-mRFP and GFP-LC3B before and after addition of GPN. Arrowheads point to lysosomes where BLTP3A and LC3B decorate the entire profile upon addition of GPN. Time, seconds. Scale bar, 5 μm. (**D**) Genomic sequence of the edited BLTP3A locus in A549 cell. Blue, gRNA; green, PAM; red, indel mutations. (**E**) Quantification of relative LAMP1 expression from western blots ($N = 3$, biological replicates) of Fig. 7A. Error bars indicate the standard error of the mean (SEM).

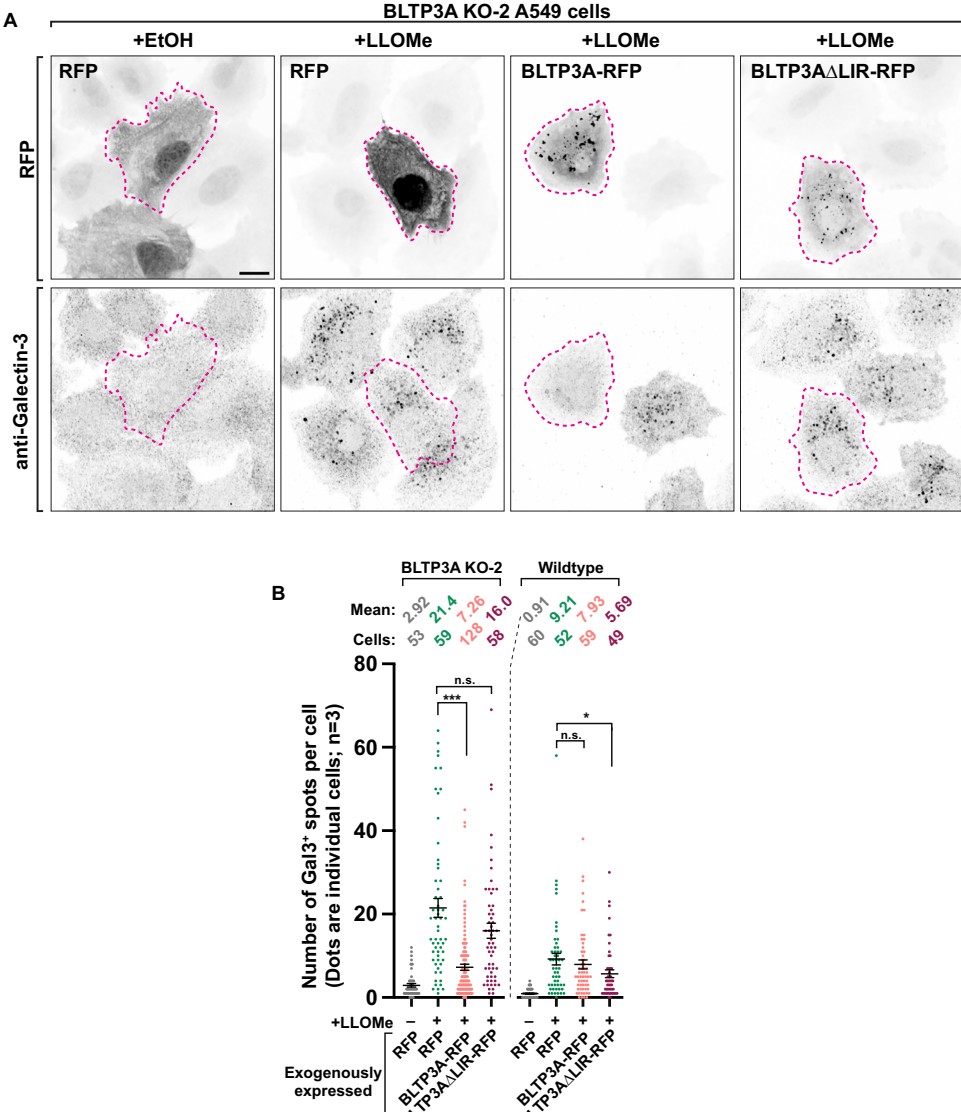

**Figure EV4. Lysosomal fragility from loss of BLTP3A is rescued by BLTP3A over-expression in A549 cells.**

(**A**) Fluorescence images (inverted greys) of BLTP3A KO A549 cells expressing indicated RFP protein (top row) with antibodies against galectin-3 (bottom row). Dotted magenta line indicates cell boundary. Cells were treated with vehicle control (left column) or 1 mM LLOMe (right three columns). Scale bar, 10 μm. (**B**) Quantification of galectin-3 spots per cell from field (**A**) ($N = 3$, biological replicates.). Error bars report the standard error of the mean (SEM). ***$P < 0.001$; **$P < 0.01$; *$P < 0.05$; n.s., not significant. Mean number of galectin-3 spots per cell and number of cells counted per condition indicated.

