## [Peer Review File · The EMBO Journal]

BLTP3A is associated with membranes of the late endocytic pathway and is an effector of CASM

Michael Hanna, Hely Rodriguez Cruz, Kenshiro Fujise, Yumei Wu, C.Shan Xu, Song Pang, Zhuonging Li, Mara Monetti, and Pietro De Camilli

Corresponding author(s): Pietro De Camilli (pietro.decamilli@yale.edu) , Michael Hanna (michael.hanna@yale.edu)

Review Timeline:

Submission Date:	3rd Apr 25
Editorial Decision:	2nd May 25
Revision Received:	22nd Jun 25
Editorial Decision:	14th Jul 25
Revision Received:	25th Jul 25
Accepted:	1st Aug 25

Editor: William Teale

Transaction Report:

Dear Dr. De Camilli,

Thank you again for the submission of your manuscript entitled "BLTP3A is associated with membranes of the late endocytic pathway and is an effector of CASM" and for your patience during the review process. We have now received the reports from the referees, which I copy below.

As you can see from their comments, both found the experiments well-framed and the topic timely. That said, both of them point out some complementation experiments that will require your attention before your manuscript can be published in The EMBO Journal. I would also be interested to hear whether you think that Referee #1's question about the mechanism of rapid, damage-induced BLTP3A release is currently reasonably experimentally accessible.

Based on the overall interest expressed in the reports, though, I would like to invite you to address the comments of all referees in a revised version of the manuscript. I should add that it is The EMBO Journal policy to allow only a single major round of revision and that it is therefore important to resolve the main concerns at this stage. I believe the concerns of the referees are reasonable and addressable, but please contact me if you have any questions (I am also available to Zoom at any time), need further input on the referee comments or if you anticipate any problems in addressing any of their points. Please, follow the instructions below when preparing your manuscript for resubmission.

I would also like to point out that as a matter of policy, competing manuscripts published during this period will not be taken into consideration in our assessment of the novelty presented by your study ("scooping" protection). We have extended this 'scooping protection policy' beyond the usual 3 month revision timeline to cover the period required for a full revision to address the essential experimental issues. Please contact me if you see a paper with related content published elsewhere to discuss the appropriate course of action.

Again, please contact me at any time during revision if you need any help or have further questions.

Thank you very much again for the opportunity to consider your work for publication. I look forward to your revision.

Best regards,

William Teale

William Teale, Ph.D.
Editor
The EMBO Journal

When submitting your revised manuscript, please carefully review the instructions below and include the following items:

- 1) a .docx formatted version of the manuscript text (including legends for main figures, EV figures and tables). Please make sure that the changes are highlighted to be clearly visible.
- 2) individual production quality figure files as .eps, .tif, .jpg (one file per figure).
- 3) a .docx formatted letter INCLUDING the reviewers' reports and your detailed point-by-point response to their comments. As part of the EMBO Press transparent editorial process, the point-by-point response is part of the Review Process File (RPF), which will be published alongside your paper.
- 4) a complete author checklist, which you can download from our author guidelines ([https://wol-prod-cdn.literatumonline.com/pb-assets/embo-site/Author Checklist%20-%20EMBO%20J-1561436015657.xlsx](https://wol-prod-cdn.literatumonline.com/pb-assets/embo-site/Author%20Checklist%20-%20EMBO%20J-1561436015657.xlsx)). Please insert information in the checklist that is also reflected in the manuscript. The completed author checklist will also be part of the RPF.
- 5) Please note that all corresponding authors are required to supply an ORCID ID for their name upon submission of a revised manuscript.
- 6) We require a 'Data Availability' section after the Materials and Methods. Before submitting your revision, primary datasets produced in this study need to be deposited in an appropriate public database, and the accession numbers and database listed

under 'Data Availability'. Please remember to provide a reviewer password if the datasets are not yet public (see <https://www.embopress.org/page/journal/14602075/authorguide#datadeposition>). If no data deposition in external databases is needed for this paper, please then state in this section: This study includes no data deposited in external repositories. Note that the Data Availability Section is restricted to new primary data that are part of this study.

Note - All links should resolve to a page where the data can be accessed.

8) For data quantification: please specify the name of the statistical test used to generate error bars and P values, the number (n) of independent experiments (specify technical or biological replicates) underlying each data point and the test used to calculate p-values in each figure legend. The figure legends should contain a basic description of n, P and the test applied. Graphs must include a description of the bars and the error bars (s.d., s.e.m.).

9) We would also encourage you to include the source data for figure panels that show essential data. Numerical data can be provided as individual .xls or .csv files (including a tab describing the data). For 'blots' or microscopy, uncropped images should be submitted (using a zip archive or a single pdf per main figure if multiple images need to be supplied for one panel). Additional information on source data and instruction on how to label the files are available at .

10) We replaced Supplementary Information with Expanded View (EV) Figures and Tables that are collapsible/expandable online (see examples in <https://www.embopress.org/doi/10.15252/embj.201695874>). A maximum of 5 EV Figures can be typeset. EV Figures should be cited as 'Figure EV1, Figure EV2" etc. in the text and their respective legends should be included in the main text after the legends of regular figures.

12) Our journal encourages inclusion of *data citations in the reference list* to directly cite datasets that were re-used and obtained from public databases. Data citations in the article text are distinct from normal bibliographical citations and should directly link to the database records from which the data can be accessed. In the main text, data citations are formatted as follows: "Data ref: Smith et al, 2001" or "Data ref: NCBI Sequence Read Archive PRJNA342805, 2017". In the Reference list, data citations must be labeled with "[DATASET]". A data reference must provide the database name, accession number/identifiers and a resolvable link to the landing page from which the data can be accessed at the end of the reference. Further instructions are available at .

13) In order to increase the reproducibility and reach of your work, The EMBO Journal includes a table of reagents that were used in the study. Please provide this along with your revisions.

Further instructions for preparing your revised manuscript:

We realize that it is difficult to revise to a specific deadline. In the interest of protecting the conceptual advance provided by the work, we recommend a revision within 3 months (31st Jul 2025). Please discuss the revision progress ahead of this time with the editor if you require more time to complete the revisions. Use the link below to submit your revision:

Referee #1:

The manuscript by Hanna et al. reports that the putative lipid channel protein BLTP3A associates with damaged lysosomes in a CASM-dependent manner to maintain lysosome health. The authors find that BLTP3A is associated with discrete vesicle-like structures when expressed from its endogenous locus, whereas overexpression of BLTP3A induces clusters of small vesicles, some of which are associated with lysosomes. Whereas the N-terminal region of BLTP3A was found to associate with lysosomes (via Rab7 binding), the C-terminus was found to mediate association with small vesicles positive for ATG9 and VAMP7. Surprisingly, lysosome damage with LLOMe caused rapid dissociation of BLTP3A from the lysosomes, but this was followed by its reassociation after 5-10 minutes. The latter depended on CASM and LIR-dependent binding of BLTP3A to LC3B. In BLTP3A knockout cells, lysosome recovery after LLOMe-induced damage was delayed as assessed with Gal3 staining. The authors conclude that BLTP3A is part of a CASM-dependent response that minimizes lysosome damage.

The data presented here are novel and shed further light on the mechanisms of lysosome repair, a timely topic. Conclusions are well supported by the data, which include IF microscopy, CLEM, EM, knock-out and knock-in studies, and biochemical analyses. I am sure this will be of interest for the readership of the EMBO Journal, but the authors should address the following issues:

Major points:

1. The lipid channels ATG2 and VPS13C have previously been shown to mediate lysosome repair. Why would a third (putative) channel, BLTP3A, be needed? Would the opposite orientation of BLTP3A compared with ATG2 and VPS13C imply that it transports lipids in the opposite direction? Any insight into this would improve the impact of the present work.
2. The mechanism behind the rapid dissociation of BLTP3A from lysosomes upon LLOMe-induced damage should be investigated. Could it be caused by displacement of Rab7 binding by VPS13C?
3. In order to prove that lipid transport underlies the function of BLTP3A in lysosome repair or resilience, it would be useful to perform rescue experiments with wild-type BLTP3A vs a mutant predicted to be defective in lipid transfer.

Minor point:

The BLTP3B expression data in Fig. 1C remain inconclusive as the antibody stains many bands at different of different size.

Referee #2:

This study investigated the cellular role of BLTP3A, a bridge-type lipid transfer protein. The authors showed that BLTP3A binds Rab7 on late endosome/lysosomal membranes through its N-terminal region and anchors vesicles containing VAMP7 and VAMP4 SNARE proteins via its C-terminal domain. Unexpectedly, upon lysosomal membrane damage induced by reagents like LLOMe, BLTP3A rapidly dissociated from lysosomes. By careful time-course observations, the authors found that BLTP3A reassociates with lysosomes and that this reassociation is dependent on the interaction with mATG8 family proteins (LC3, GABARAP) through a conserved LIR motif. This CASM-dependent mechanism suggests BLTP3A plays a significant role in the cellular response to lysosomal damage, supported by findings that BLTP3A KO cells exhibit increased lysosomal fragility. Thus, the authors concluded that BLTP3A is an effector of CASM and contributes to lysosomal homeostasis.

This study is highly evaluated for uncovering a novel mechanism whereby BLTP3A functions as an effector of the CASM pathway. The authors have robustly supported this conclusion through diverse and complementary methodologies, including endogenous protein tagging, genetic knockout, live-cell imaging, and CLEM. Data showing increased lysosomal fragility in BLTP3A-deficient cells suggest a critical physiological role for BLTP3A in maintaining lysosomal homeostasis. Although the precise mechanism by which BLTP3A, a bridge-type lipid transfer protein, regulates lysosomal homeostasis remains unclear and represents a limitation of the current work, the overall findings are highly valuable and relevant for a broad readership of The EMBO Journal. Therefore, the paper merits publication after addressing the following concerns, particularly the strengthening of the data presented in Figures 7C,D as written below.

Major points

1) To more conclusively demonstrate the functional importance of BLTP3A in lysosomal repair, the authors should perform rescue experiments by reintroducing wild-type BLTP3A into BLTP3A knockout cells and testing whether this expression alleviates lysosomal damage.

2) Using the same experimental system, the authors should also assess the impact on lysosomal damage of expressing BLTP3A mutants lacking either the LIR motif or the C-terminal region responsible for vesicle tethering. These experiments will clarify whether BLTP3A-mediated lysosomal repair relies on the CASM pathway and/or vesicle tethering activity. If the latter proves critical, the authors should discuss the possibility that tethered membrane vesicles serve as a source of membranes for lysosomal repair.

Minor point

In Figure 1H, the fluorescence image does not seem to accurately correspond with the EM image. Ensure that exactly the same region is displayed.

Response to the review. (*Authors responses are italicized*)

We would like to thank both reviewers for the many positive comments. We found their suggestions very helpful. Our point-by-point responses to their comments, which also take into account the editorial letter are appended below.

While revising the paper we realized that in the original submission we did not include as supplemental material some movies related to galleries of images shown in the main figures (Figure 4A, 4B, 5C, 5D, 6D, EV3C). The information contained in these movies is already present in the galleries of still images, but the movies make the message even more clear. These are presented now as EV movies 1-7 & 10.

Detailed point-to point response to the reviewers' critiques:

Referee #1:

The manuscript by Hanna et al. reports that the putative lipid channel protein BLTP3A associates with damaged lysosomes in a CASM-dependent manner to maintain lysosome health. The authors find that BLTP3A is associated with discrete vesicle-like structures when expressed from its endogenous locus, whereas overexpression of BLTP3A induces clusters of small vesicles, some of which are associated with lysosomes. Whereas the N-terminal region of BLTP3A was found to associate with lysosomes (via Rab7 binding), the C-terminus was found to mediate association with small vesicles positive for ATG9 and VAMP7. Surprisingly, lysosome damage with LLOMe caused rapid dissociation of BLTP3A from the lysosomes, but this was followed by its reassociation after 5-10 minutes. The latter depended on CASM and LIR-dependent binding of BLTP3A to LC3B. In BLTP3A knockout cells, lysosome recovery after LLOMe-induced damage was delayed as assessed with Gal3 staining. The authors conclude that BLTP3A is part of a CASM-dependent response that minimizes lysosome damage.

The data presented here are novel and shed further light on the mechanisms of lysosome repair, a timely topic. Conclusions are well supported by the data, which include IF microscopy, CLEM, EM, knock-out and knock-in studies, and biochemical analyses. I am sure this will be of interest for the readership of the EMBO Journal, but the authors should address the following issues:

We would like to thank the reviewer for their thoughtful and pointed analysis of our manuscript. We were pleased to read that they found our study "novel" and that it sheds "further light on the mechanisms of lysosome repair, a timely topic."

Major points:

1. The lipid channels ATG2 and VPS13C have previously been shown to mediate lysosome repair. Why would a third (putative) channel, BLTP3A, be needed? Would the opposite orientation of BLTP3A compared with ATG2 and VPS13C imply that it transports lipids in the opposite direction? Any insight into this would improve the impact of the present work.

We can only speculate on why multiple lipid transport proteins are needed. Some speculations are listed below, but we would prefer not to expand beyond what we have already included in the manuscript.

1) Timing of recruitment as a key difference: VPS13C is rapidly (within 10s of seconds) recruited to the surface of lysosomes upon addition of the damaging agent (by recognizing membrane packing defects, PMID: 40211074), while BLTP3A is recruited later (5-10 mins after addition of the damaging agent) as an effector of CASM. Different recruiting mechanisms may or may not be activated depending on the extent of lysosome damage.

2) Differences in protein expression between tissues: VPS13C and ATG2 proteins are expressed ubiquitously, while expression of BLTP3A is heterogeneous, with the highest expression in brain and lung. The function of BLTP3A may therefore be especially important in some cells/tissues.

Concerning the question of the orientation of BLTP3A at the lysosome surface: BLTPs are thought to unidirectionally transfer lipids at any given time due to a bottleneck in the hydrophobic channel where lipids can only flow in single file. What controls the direction of transfer (i.e. N to C terminus or vice versa) is not understood and it remains unclear whether they can transfer lipids in either direction but at different times. The reviewer points out a different orientation of BLTP3A at the lysosome surface relative to that reported for VPS13C and ATG2. However, this is the situation before lysosome damage. After lysosome damage, the recruitment of BLTP3A via an interaction between its LIR motif within a large disordered outpocketing and ATG8 proteins on the surface of lysosomes would allow for an orientation consistent with those of ATG2 and VPS13C. Identifying other interacting partners that may direct the orientation of BLTP3A at damaged lysosomes is a very high priority of our ongoing efforts.

2. The mechanism behind the rapid dissociation of BLTP3A from lysosomes upon LLOMe-induced damage should be investigated. Could it be caused by displacement of Rab7 binding by VPS13C?

Thank you for raising the importance of understanding the dissociation of BLTP3A from lysosomes upon damage. In fact we had already attempted to address this issue (in the original manuscript) by testing the role of Rab7 phosphorylation on the loss of BLTP3A from the surface of lysosomes in response to damage. Rab7 (S72) is phosphorylated rapidly in response to lysosome damage by LRRK1 (PMID: 33459343). We hypothesized that the association of BLTP3A with Rab7 could be affected by phosphorylation as demonstrated for other Rab7 effectors (PMID: 38728007). To test for this possibility we had expressed in RPE-1 cells BLTP3A-mRFP with either WT or kinase-active (KA) GFP-LRRK1 (see Fig. EV2C-D). Expression of LRRK1_{KA} resulted in robust Rab7 phosphorylation, but no change in the localization of BLTP3A-mRFP to lysosomes, suggesting that BLTP3A associates with Rab7 independently of phosphorylation at S72.

We agree that it is possible that a Rab7 interacting partner, such as VPS13C, which is rapidly recruited to the surface of damaged lysosomes, may displace BLTP3A. Therefore, we have now utilized VPS13C knock-out A549 cells to test this idea. BLTP3A-1-336-mRFP (a fragment of BLTP3A which specifically binds to lysosomes, but not vesicles, in a Rab7 dependent manner – see Fig. 2E and Fig. 4B) localized with LAMP1-GFP structures in both WT and VPS13C KO A549 cells. Upon the addition of LLOMe in either WT or VPS13C KO cells, we observed that BLTP3A-1-336-mRFP became cytosolic, no longer localizing with LAMP1-GFP. As VPS13C is not present in these cells, these new data suggest that VPS13C is not sufficient to displace a BLTP3A from lysosomes in response to damage. This data is presented in **Fig. EV2A-B**.

3. In order to prove that lipid transport underlies the function of BLTP3A in lysosome repair or resilience, it would be useful to perform rescue experiments with wild-type BLTP3A vs a mutant predicted to be defective in lipid transfer.

We agree with the reviewer regarding the importance of defining the role of lipid transfer via BLTP3A on lysosome fragility. To address this comment, we engineered mutations of hydrophobic residues that line the floor of the channel of BLTP3A. Residues were selected based on previous publications generating putative 'lipid transfer dead mutants' (LTDM) of other BLTPs (PMID: 32182622; 37645754). Unfortunately, the incorporation of these mutations led to the significant mislocalization of exogenous BLTP3A, which appeared to be ectopically localized on or near mitochondria (see figure below). Therefore, we did not proceed to use these BLTP3A mutants in rescue experiments.

Figure legend. Fluorescence microscopy images of RPE-1 cells expressing predicted lipid transfer dead mutants (LTDM1, left; LTDM2, right) BLTP3A-RFP (magenta) and organelle markers of lysosomes (NPC1-GFP; green) and mitochondria (mito-BFP; blue). Scale bar, 5 μ m. Higher magnification zooms show individual channels in inverted greys. Merge of high mag reveals ectopic localization of mutant BLTP3A-RFP with mitochondria and no accumulations on the surface of lysosomes as observed with wild-type BLTP3A (see Figure 1G in manuscript). Scale bar, 1 μ m. **LTDM1 mutations:** C63R, A66D, L84R, V89E, I127D, I132K, F145E, L150E, W185R, I190K, I216D. **LTDM2 mutations:** C63K, L82K, V87E, V129E, I134R, F145E, L150E, I183D, L188K, L209E, I216R

*However, we have now performed rescue experiments using wild-type BLTP3A or BLTP3A lacking the LIR motif (BLTP3A Δ LIR). The number of Gal3 spots (which reflect lysosome damage) per cell in BLTP3A KO cells expressing wild-type BLTP3A, but not BLTP3A Δ LIR, was similar to that measured in wild-type cells. This data is presented in **Fig. EV4A-B**.*

Minor point:

The BLTP3B expression data in Fig. 1C remain inconclusive as the antibody stains many bands at different of different size.

As data on BLTP3B expression are not needed for this study, we have removed the BLTP3B blot and modified the following sentence accordingly “As revealed by western blotting, BLTP3A, ~~like BLTP3B~~, has broad expression in different mouse tissues (Figure 1C), with higher levels occurring in brain and lung”

Referee #2:

This study investigated the cellular role of BLTP3A, a bridge-type lipid transfer protein. The authors showed that BLTP3A binds Rab7 on late endosome/lysosomal membranes through its N-terminal region and anchors vesicles containing VAMP7 and VAMP4 SNARE proteins via its C-terminal domain. Unexpectedly, upon lysosomal membrane damage induced by reagents like LLOMe, BLTP3A rapidly dissociated from lysosomes. By careful time-course observations, the authors found that BLTP3A reassociates with lysosomes and that this reassociation is dependent on the interaction with mATG8 family proteins (LC3, GABARAP) through a conserved LIR motif. This CASM-dependent mechanism suggests BLTP3A plays a significant role in the cellular response to lysosomal damage, supported by findings that BLTP3A KO cells exhibit increased lysosomal fragility. Thus, the authors concluded that BLTP3A is an effector of CASM and contributes to lysosomal homeostasis.

This study is highly evaluated for uncovering a novel mechanism whereby BLTP3A functions as an effector of the CASM pathway. The authors have robustly supported this conclusion through diverse and complementary methodologies, including endogenous protein tagging, genetic knockout, live-cell imaging, and CLEM. Data showing increased lysosomal fragility in BLTP3A-deficient cells suggest a critical physiological role for BLTP3A in maintaining lysosomal homeostasis. Although the precise mechanism by which BLTP3A, a bridge-type lipid transfer protein, regulates lysosomal homeostasis remains unclear and represents a limitation of the current work, the overall findings are highly valuable and relevant for a broad readership of The EMBO Journal. Therefore, the paper merits publication after addressing the following concerns, particularly the strengthening of the data presented in Figures 7C,D as written below.

We would like to thank the reviewer for their careful and thoughtful analysis of our manuscript. We were pleased to read that our study is “highly evaluated for uncovering a novel mechanism whereby BLTP3A functions as an effector of the CASM pathway” and that we have “robustly supported this conclusion.”

Major points

1) To more conclusively demonstrate the functional importance of BLTP3A in lysosomal repair, the authors should perform rescue experiments by reintroducing wild-type BLTP3A into BLTP3A knockout cells and testing whether this expression alleviates lysosomal damage.

*We have now performed rescue experiments as suggested to assess the impact of the lack of BLTP3A on lysosome fragility (using the Gal3 assay) in BLTP3A KO cells, finding that rescues occur by expressing wild-type BLTP3A-mRFP, but not RFP alone. This data is presented in **Fig. EV4A-B**.*

2) Using the same experimental system, the authors should also assess the impact on lysosomal damage of expressing BLTP3A mutants lacking either the LIR motif or the C-terminal region responsible for vesicle tethering. These experiments will clarify whether BLTP3A-mediated lysosomal repair relies on the CASM pathway and/or vesicle tethering activity. If the latter proves critical, the authors should discuss the possibility that tethered membrane vesicles serve as a source of membranes for lysosomal repair.

*In addition to the rescue experiments described above (major point #1), and in response to this comment, we also performed rescue experiments using a construct of BLTP3A that lacks the LIR motif (BLTP3A Δ LIR-mRFP) and found that this motif is required for the rescue. This data is presented in **Fig. EV4A-B**.*

We decided against using the BLTP3A vesicle binding mutant (BLTP3A-1-1327-mRFP) for rescue experiments as this truncation, while disrupting the ability of BLTP3A to associate with vesicles, also lacks the final two β -strands of the hydrophobic channel. This would likely affect its ability to transfer lipids and make interpreting rescue, or failure to rescue, difficult. Defining the association between BLTP3A and vesicles is a very high priority of our immediate future aims. Concerning the possibility that the small BLTP3A vesicles may contribute lipids to the lysosomes by a bridge-like mechanism, we find this unlikely given their already very small size.

Minor point

In Figure 1H, the fluorescence image does not seem to accurately correspond with the EM image. Ensure that exactly the same region is displayed.

Thank you for pointing this out. The fluorescence image is correct, but the alignment is ajar. We have now corrected the alignment.

Dear Dr. De Camilli,

We sent your revised manuscript and point-by-point response to both referees and have now received a brief report from one of them, which I have included below. As you will see, you have addressed all concerns satisfactorily. Before I can finally accept the manuscript, there are some remaining editorial points which need to be addressed. In this regard would you please:

- update the email address for Michael Hanna in the manuscript text,
 - check the email address for Kenshiro Fujise (kenshiro.fujise@yale.edu) is still valid,
 - include the "Funding" section along with acknowledgements "Acknowledgements",
 - limit keywords to a total of five, and list beneath the abstract,
 - for longer author lists in the reference section, list the first ten names and then "+ et al.",
 - rename the conflict of interests statement as the "Disclosure and competing interests statement",
 - remove the AC/Credit section from the text,
 - source file names, titles, legends and manuscript callouts all need to be updated to Dataset EV1-EV2 instead of Table EV1-EV2, their legends should be removed from the manuscript and uploaded as a separate tab/sheet in each Excel file,
 - remove the Reagents and Tools table from the manuscript text and upload as an individual file using the template from our guide to authors,
 - change the name of the "Materials and Methods" section to "Methods"
- rename movie files as Movie EV1-EV10 with the corresponding callouts in the manuscript text; remove the legends from the manuscript text and zip with each movie file, and
- use the section order: Title page - Abstract & Keywords - Introduction - Results - Discussion - Methods - Data Availability - Acknowledgements - Disclosure and Competing Interests Statement - References - Figure Legends - Table(s) - Expanded View Figure Legends.

We include a synopsis of the paper (see <http://emboj.embopress.org/>). Please provide me with a general summary image, a two sentence statement and 3-5 bullet points that capture the key findings of the paper.

I am looking forward to receiving your revised manuscript.

EMBO Press is an editorially independent publishing platform for the development of EMBO scientific publications.

Best wishes,

William Teale

William Teale, PhD
Editor
The EMBO Journal
w.teale@embojournal.org

- a Reagents and Tools Table as part of the Methods section, which can be downloaded from our author guidelines

(<https://www.embopress.org/page/journal/14602075/authorguide#structuredmethods>)

We realize that it is difficult to revise to a specific deadline. In the interest of protecting the conceptual advance provided by the work, we recommend a revision within 3 months (12th Oct 2025). Please discuss the revision progress ahead of this time with the editor if you require more time to complete the revisions. Use the link below to submit your revision:

Referee #2:

The authors have adequately addressed my concerns, and the manuscript has sufficiently improved.

All editorial and formatting issues were resolved by the authors.

Dear Pietro,

I am pleased to inform you that your manuscript has been accepted for publication in the EMBO Journal.

Congratulations to you and your team! I am excited about seeing this article in the EMBO Journal.

Best wishes,

William

William Teale, PhD
Editor
The EMBO Journal
w.teale@embojournal.org
